# Reviews and syntheses: On the roles trees play in building and plumbing the critical zone

Susan L. Brantley[1], David M. Eissenstat[2], Jill A. Marshall[3,4], Sarah E. Godsey[5], Zsuzsanna Balogh-Brunstad[6], Diana L. Karwan[7], Shirley A. Papuga[8,9], Joshua Roering[10], Todd E. Dawson[11], Jaivime Evaristo[12], Oliver Chadwick[13], Jeffrey J. McDonnell[14], Kathleen C. Weathers[15]

[1]Earth and Environmental Systems Institute and Department of Geosciences, Pennsylvania State University, PA, USA
[2]Department of Ecosystem Science and Management, Pennsylvania State University, PA, USA
[3]Earth and Planetary Science, University of California-Berkeley, Berkeley, CA, USA
[4]Institute of Alpine and Arctic Research (INSTAAR), University of Colorado, Boulder, Colorado 80309, USA
[5]Department of Geosciences, Idaho State University, Pocatello, ID, USA
[6]Department of Geology and Environmental Sciences, Hartwick College, Oneonta, NY, USA
[7]Department of Forest Resources, University of Minnesota, Saint Paul, MN, USA
[8]School of Natural Resources and Environment, University of Arizona, Tucson, AZ, USA
[9] Department of Geology, Wayne State University, Detroit, MI, USA
[10]Department of Geological Sciences, University of Oregon, Eugene, OR, USA
[11]Department of Integrative Biology, University of California, Berkeley, CA, USA
[12]Department of Natural Resources and Environmental Science, University of Nevada, Reno, Reno, Nevada, USA
[13]Department of Geography, University of California-Santa Barbara, Santa Barbara, CA, USA
[14]School of Environment and Sustainability, University of Saskatchewan, Saskatoon, Canada
[15]Cary Institute of Ecosystem Studies, Millbrook, NY, USA

*Correspondence to*: Susan L. Brantley (brantley@eesi.psu.edu)

**Abstract.** Trees, the most successful biological power plants on earth, build and plumb the critical zone (CZ) in ways that we do not yet understand. To encourage exploration of the character and implications of interactions between trees and soil in the CZ, we propose nine hypotheses that can be tested at diverse settings. The hypotheses are roughly divided into those about the architecture (building) and those about the water (plumbing) in the critical zone, but the two functions are intertwined. Depending upon one's disciplinary background, many of the nine hypotheses listed below may appear obviously true or obviously false. 1) Tree roots can only physically penetrate and biogeochemically comminute the immobile substrate underlying mobile soil where that underlying substrate is fractured or pre-weathered. 2) In settings where the thickness of weathered material, $H$, is large, trees primarily shape the CZ through biogeochemical reactions within the rooting zone. 3) In forested uplands, the thickness of mobile soil, $h$, can evolve toward a steady state because of feedbacks related to root disruption and tree throw. 4) In settings where $h << H$ and the rate of uplift and erosion are low, the uptake of phosphorus into trees is buffered by the fine-grained fraction of the soil, and the ultimate source of this phosphorus is dust. 5) In settings of limited water availability, trees maintain the highest length density of functional roots at depths where water can be extracted over most of the growing season with the least amount of energy expenditure. 6) Trees grow the majority of their roots in the zone where the most growth-limiting resource is abundant, but they also grow roots at other depths to forage for other resources and to hydraulically redistribute those resources to depths where they can be taken up more efficiently. 7) Trees rely on matrix

water in the unsaturated zone that at times may have an isotopic composition distinct from the gravity-drained water that transits from the hillslope to groundwater and streamflow. 8) Mycorrhizal fungi can use matrix water directly but trees can only use this water by accessing it indirectly through the fungi. 9) Even trees growing well above the valley floor of a catchment can directly affect stream chemistry where changes in permeability near the rooting zone promote intermittent zones of water saturation and downslope flow of water to the stream. By testing these nine hypotheses, we will generate important new cross-disciplinary insights that advance critical zone science.

## 1 Introduction

Natural scientists have long known that soils affect biota and biota affect soils (e.g. Belt, 1874). The perspective most commonly invoked by soil scientists to study such phenomena emphasizes timescales from years to centuries and depths from centimeters to meters (e.g. Dokuchaev, 1883). By contrast, geologists commonly study soil and other altered material to depths as large as 1000s of meters over timeframes as long as millions of years (e.g. Becker, 1895; Ollier, 1984). Now, a new field of science bridges these depth and temporal differences in perspective by targeting the entire weathering engine from vegetation canopy to deep bedrock and by developing quantitative models for the evolution and dynamics of the landscape. This zone has been named the "critical zone" (CZ), given its importance to life on this planet (U.S. National Research Council Committee on Basic Research Opportunities in the Earth Sciences, 2001). Implicit to CZ science is the idea that investigating both the abiotic and biotic CZ over all relevant timescales and depths will elucidate the form and function of the CZ itself and allow projections of its future forms and functions. One CZ focus is organismal. As such, a specific focus is on trees – the most successful terrestrial entities transforming solar energy into the chemical energy of biomass. While many researchers have investigated the effect of vegetation on soils and weathering (e.g., Berner et al., 2003; Brantley et al., 2012), the emphasis of CZ science on deeper processes demands a focus on organisms such as trees that impact regolith over greater depths. In this paper, we highlight some puzzles about the nature of trees' effect on the CZ and the CZ's effect on trees.

Like industrial power plants, trees cycle large volumes of water as they transform the energy of the sun into chemical energy (Figure 1): estimates based on isotope measurements suggest that 50 to 65% of the incoming solar energy used by trees during growth moves water through vascular tissues from roots to leaves through transpiration (Jasechko et al., 2013; Schlesinger and Jasechko, 2014). In addition to moving hydrogen and oxygen, trees move 16 essential nutrients from the soil and rock into biomass along with 14 or so other less essential micronutrients (Sterner and Elser, 2002; Cornelis et al., 2009). At the same time, trees fix carbon from the

atmosphere into carbohydrates which are moved in the tree's phloem tissues. As trees cycle water and nutrients (Fig. 1), they also enrich parts of the soil with these nutrients. As biotic engines, trees thus strongly impact the energy, water, and element cycles in forested and savannah ecosystems, shaping and sculpting landscapes and soils over long timescales (Reneau and Dietrich, 1991; van Breemen et al., 2000; Balogh-Brunstad et al., 2008a; Pawlik et al., 2016). Soils and landscapes in turn affect plant species composition and size as well as above- and below-ground productivity and rooting depth (Bennie, 1991; Clark et al., 2003; Hahm et al., 2014; Marshall and Roering, 2014). Only by studying the entire CZ using concepts from hydrology, soil science, geomorphology, geochemistry, and ecology will a synthetic view of tree-soil-landscape co-evolution emerge. Here, we promote the emergence of this new understanding by posing nine hypotheses about trees as builders and plumbers of the CZ (Figure 1).

These hypotheses were crafted to target some of the key points that puzzle us and that warrant further research. Some holes in our understanding are obvious. For example, many numerical models are available to simulate chemical weathering and erosion (Lichtner, 1988; Lebedeva et al., 2007; Minasny et al., 2008; Maher et al., 2009) but most only model trees indirectly by incorporating the assumption that trees can reduce the water flow through the soil through evapotranspiration. Where the impact of trees or biota have been incorporated into models of weathering or landscape development, the models typically focus on one aspect of trees' impact (Gabet and Mudd, 2010; Roering et al., 2010; Corenblit et al., 2011; Reinhardt et al., 2011; Godderis and Brantley, 2014). Many of our hypotheses target these holes in our understanding.

We also identified hypotheses that have arisen because we now can measure new phenomena, new hydrologic or chemical reservoirs, or new types of microbiota. For example, it is obvious that the water in many streams derives from rainfall. Yet other research suggests that the water that trees use might be different from water that flows into streams (Brooks et al., 2010; Evaristo et al., 2015). Indeed, all along the path of waterflow from the atmosphere to streams, trees act as valves that re-direct water (Fig. 1). For example, the first "valve" is the canopy: as rainfall enters the canopy, some water is retained (interception) and some falls directly to the soil (throughfall). The intercepted water is in turn re-evaporated back to the atmosphere or may pass through the network of leaves and branches with some flowing down the tree trunk (stemflow). This stemflow typically contains nutrients derived from dust and foliar leaching, and these nutrients are delivered to the subsurface as flow down the trunk and along the roots, spreading out, and sometimes reaching deep into the soil profile beneath the tree. This collection throughout the canopy and re-distribution of water throughout the root network has been described as 'double-funneling' (Johnson and Lehmann, 2006). While some of this water flows downward beneath the tree, some flows

laterally along roots and their associated macropores at shallower depths (Newman et al., 2004). In addition to downward and lateral flow in the subsurface, in the early 1990s it was hypothesized that trees could lift water from depth up to the surface (hydraulic lift); it was eventually shown that trees can pump water both upward and downward (hydraulic re-distribution) through the soil (Burgess et al., 1998). Movement of water by the tree in turn

results in development of a heterogeneous distribution of nutrients, soil pH, cation exchange capacity, soil organic carbon, and micro-organisms (Johnson and Lehmann, 2006).

These observations point out that there is a generally uncharacterized heterogeneity of water resources, nutrients, and fluxes in the CZ related to trees (Johnson and Lehmann, 2006; Oshun et al., 2016; Bowling et al., 2017). These

findings are now forcing researchers to develop new ways to investigate the parts of the CZ that trees access. In turn, this is driving a new re-calculation of the types, sizes, and residence times of water inventories that are available to plants in catchments (Oshun et al., 2016) and how water use is changing with atmospheric carbon content (Keenan et al., 2013). We also know that nearly all tree species host mycorrhizal fungi in symbiotic association with their roots (Read, 1997). However, our understanding of the roles these fungi play in CZ processes

is in its infancy. Some reports suggest that up to a third of the organic material formed during photosynthesis by trees is exchanged with mycorrhizal fungi for nutrients and water (Read, 1997; Leake et al., 2008). Since the surface area to volume ratio of fungal hyphae that absorb soil-borne resources far exceeds that same ratio for tree roots, mycorrhizal fungi are a key player in building and plumbing the CZ.

The paper begins with summary sections about evolution and distribution of tree roots and fungi, and a section on the structure of the CZ itself. Table 1 summarizes the nomenclature we use. Such terminology is inherently difficult because we use it to describe somewhat operationally-defined and arbitrary layers and types of water in the CZ whereas both the soil and the water exist across gradients rather than within strict delineated compartments. The rest of the paper consists of two sections on building and plumbing the critical zone that respectively contain four

and five hypotheses each. Trees *build* the CZ by altering the physical architecture and chemistry of the subsurface environment. Trees *plumb* the CZ because they impact the reservoirs, pathways, and fluxes of water in the subsurface. The two subsets of hypotheses that focus on building and plumbing the CZ each highlight processes with inherently different characteristic timescales. In the first section of the paper, we pose questions about how trees affect the CZ architecture and we thus focus on questions related to processes that steer solute and sediment

production and erosion over timescales of decades to millenia. In the second part of the paper, we focus on how trees affect the movement of water at timescales of seconds to decades. This water passes through the architecture

described in part 1, facilitating chemical, physical, and biological interactions. Of course, this distinction into building and plumbing is itself arbitrary and in many cases both functions are intertwined, and this concept is discussed in a synthesis section at the end of the paper.

We designed the paper to highlight areas of contradiction among disciplines and to clarify the new hypotheses that are emerging within the cross-disciplinary dialogue in CZ science. The paper thus provides a roadmap of puzzles to stimulate the research of the future.

**1.1 Evolution of tree-fungi interactions**

In addition to growing roots to anchor the tree, plants grow roots to take up water and nutrients and consume oxygen
and carbohydrates to support the metabolism required for these functions (Stewart et al., 1999). As noted above, most tree roots are associated with symbiotic mycorrhizal fungi (Read, 1997). The term "mycorrhiza" refers to the symbiotic association of a root ("rhiza") and a fungus ("myco"). The oldest type of such fungi, arbuscular mycorrhizal fungi (AMF), form associations with plants that are inside the cell and are thus known as endomycorrhizal (Table 1). AMF were present when plants first colonized the land surface using modified stems
before "true" roots evolved (Brundrett, 2002). As the first true roots of terrestrial vascular plants evolved, they were relatively thick and required AMF for the plant to survive (i.e., obligate association). Eventually, certain lineages of trees evolved thin roots and became facultatively associated with AM fungi: in other words, the trees could survive with or without the fungi.

These latter thin roots can readily proliferate into zones of high nutrient or water content (Adams et al., 2013; Eissenstat et al., 2015). Species with these roots can also readily allow the roots to die off if zones become barren. These late-to-evolve, thin-root species often depend less on mycorrhizas than the early-to-evolve, thick-root species. Thin roots presumably evolved to access environments unfavorable for thick roots, such as very dry soils (Chen et al., 2013). In addition to evolution of thin roots, a new type of mycorrhizal fungi known as ectomycorrhizal
fungi (EMF) evolved (Table 1). EMF do not colonize the inside of plant root cells. Specifically, in boreal and north temperate regions and other locations where nutrients often are retained in slowly decomposing organic matter, some lineages of higher fungi that were previously free-living saprotrophs (organisms utilizing non-living organic materials for food) evolved symbiotic associations with plants. These ectomycorrhizal fungi co-evolved with and fine-tuned their relationship with plants. EMF differ from AMF in that they can develop large mycelial networks
that explore large volumes of soil for water and nutrients. Today, ectomycorrhizal trees often have short, numerous

root tips that promote EMF colonization (Brundrett, 2002). In addition, EM fungi often have retained some of the enzymes associated with saprotrophs. Therefore, EM trees often are more adept than AM trees at utilizing nutrients that are organically-bound. It is also likely that the leaves of EM trees co-evolved with the EM fungi. Specifically, EM trees tend to have chemically more recalcitrant leaves that decompose less readily than those of AM trees
(Phillips et al., 2013; Lin et al., 2017).

Given the evolutionary history, two predominant characteristics determine much about the strategies that trees use to forage for water and nutrients in the soil: the thickness of the roots and the type of fungi present (Chen et al., 2016; Cheng et al., 2016). First, thin-root tree species grow roots opportunistically to search for and take up
nutrients, especially from organic-rich zones. In contrast, thick-root tree species do not show opportunistic root growth and thus rely more on their mycorrhizal fungal hyphae to explore and take up nutrients. Second, EM tree species favor foraging with their fungal hyphae rather than their roots. Thus, trees colonized by AM fungi generally forage for nutrients using their roots, especially if they have thin roots, but trees colonized by EM fungi forage more with their fungal hyphae, especially if they have thick roots.

Today, trees can have thick or thin roots and can be colonized by AM, EM, or no fungi at all. Examples of trees growing today with these characteristics include elms and maples (thin roots colonized by AMF), magnolia and tulip poplar (thick roots colonized by AMF), birches, hickories, and oaks (thin roots colonized by EMF) and species in the pine family including spruce, pines, and hemlock (thick roots colonized by EMF). Thick-root AM species
often compete best in locations with more stable nutrient availability and higher moisture conditions. In contrast, thin-root AM species are generally better at taking advantage of temporally dynamic water and nutrient conditions (Chen et al., 2013). EM species are often found in conditions where nutrients are less available and more bound in organic matter. Valley floors in temperate forests may often have more AM trees, and this is often the most common location of thick root species like tulip poplar and magnolia (Smith et al., 2017). In contrast, Smith et al.
observed that ridgetops and steep midslopes with thin soils may be colonized by EM trees or AM trees with thin roots like maples with the EM trees such as oaks often more successful on drier locations (e.g., sun-facing aspects).

**1.2 Form, function, and distribution of tree roots**

As discussed in the last section, much of the interplay between trees and earth materials is mediated by roots and their associated fungal hyphae. It is therefore important to understand where tree roots are found. In general, most
tree roots, and a very high fraction of fine roots (i.e., < 2 mm), are observed in the upper 30 cm (Schenk and

Jackson, 2005) and this upper layer is thus often referred to as the rooting zone. Indeed, almost all roots are typically located within 2 m of land surface. However, the specific depths to which tree roots penetrate vary with precipitation, potential evapotranspiration, and tree species (Gale and Grigal, 1987; Schenk and Jackson, 2002a, b). The depth of root penetration also varies with the thickness and properties of soil, and the characteristics of bedrock (Kochenderfer, 1973; Stone and Kalisz, 1991; Anderson et al., 1995; Sternberg et al., 1996; Hubbert et al., 2001; Hubbert et al., 2001a; Witty et al., 2003; Bornyasz et al., 2005; Nicoll et al., 2006; Graham et al., 2010).

In general, researchers have observed that most root mass is found in the disaggregated material above bedrock. However, where soils are shallow, the underlying substrate may contain roots, sometimes to many meters depth, especially in upland areas  (Hellmers et al., 1955; Scholl, 1976; Stone and Kalisz, 1991; Anderson et al., 1995; Canadell and Zedler, 1995; Jackson et al., 1999; Hubbert et al., 2001; Hubbert et al., 2001a; Egerton-Warburton et al., 2003; Rose et al., 2003; Witty et al., 2003; Bornyasz et al., 2005; Graham et al., 2010; Roering et al., 2010; Estrada-Medina et al., 2013). Both fine, absorptive roots and larger framework roots have been found at tens of meters depth beneath the land surface (Canadell et al., 1996; Jackson et al., 1999).

These different thicknesses of roots at depth point to the important fact that all roots are not the same, even at birth, and the type of root is important in terms of both plumbing and building the CZ.  Most roots arise from the pericycle (active dividing cells or meristemic tissue inside the root cortex) of another root. Most of the roots that form are thin and small and absorptive in nature. However, another type of larger-diameter root arising from the pericycle – commonly referred to as a pioneer root – extends rapidly and undergoes woody secondary development within weeks (Zadworny and Eissenstat, 2011).  These roots typically are not mycorrhizal and are chiefly used for transport and for building the framework of the root system. Therefore, they are generally referred to as "framework" or "woody" roots upon maturation.  While important in the root framework, such roots comprise only a very small fraction of total root length: most of the root length is derived from fine laterals that may branch two or three orders (McCormack et al., 2015).  These laterals chiefly have an absorptive function and are characterized by a relatively high nitrogen concentration. They can be colonized by mycorrhizal fungi and generally are ephemeral, living typically 0.5 to 2 years.

Most of our knowledge of deep root growth has arisen from studies in arid or semi-arid climates where water is a limiting resource. In those environments, trees must grow deep roots to harvest water in fractured or porous bedrock material (Lewis and Burgy, 1964; Zwieniecki and Newton, 1995; Hubbert et al., 2001; Hubbert et al., 2001a;

Egerton-Warburton et al., 2003; Rose et al., 2003; Witty et al., 2003; Bornyasz et al., 2005; Schenk, 2008; Graham et al., 2010; Schwinning, 2010). In contrast, in temperate regions with higher rainfall (e.g. Gaines et al., 2016), trees have been observed to access water from predominantly the upper soil even though their roots can still reach depths of several meters. In general, however, the extent of deep root penetration has not been systematically explored since most researchers have focused only on shallow depths (Maeght et al., 2013) and only a few lithologies: e.g., granite (Hubbert et al., 2001; Hubbert et al., 2001a; Witty et al., 2003; Bornyasz et al., 2005; Graham et al., 2010; Poot et al., 2012); shale (Hasenmueller et al., 2017); or limestone (Hasselquist et al., 2010; Estrada-Medina et al., 2013). For example, Hasenmueller et al. (2017) identified fine roots that penetrate meters into bedrock in a temperate humid forest where trees generally are not water limited. In the same general region, however, roots at tens of meters depth are sometimes observed in karst lithologies. The utility of deep roots in such humid forests has not been established. In temperate climates, it is possible that such deep roots allow water uptake late in the growing season when water has been depleted from shallow zones (Fimmen et al., 2007) or during drought episodes that may occur at decadal time scales. In addition to providing water access, roots at depths deeper than 20 cm may also provide access to nutrients such as Ca that are low in abundance in shallower soils. For example, roots may pump Ca into shallow soil layers for easier uptake by surficial roots (Dijkstra and Smits, 2002). Deep roots also deposit organic reducing agents in the B or C horizons that allow extraction of nutrients through Fe-C cycling (Fimmen et al., 2007).

**1.3 Architectural layering of the critical zone**

A diverse array of observations implies that trees play a significant role in building and plumbing the CZ architecture (Johnson and Lehmann, 2006; Pawlik et al., 2016). For example, paleosols and sedimentary deposits have been used to argue that clay enrichment and chemical weathering was promoted by the proliferation of forest ecosystems during the Devonian, prompting the decline of atmospheric carbon dioxide and global cooling (e.g. Retallack, 1997). Other long-term studies that relate biogeochemistry to climate have also been used to argue that tree-CZ interactions may be central to our understanding of global change (Berner et al., 2003; Taylor et al., 2009). It is also well known that trees use many mechanisms that modulate CZ processes and development (Amundson, 2004; Brantley et al., 2012). To be specific, trees have the ability to alter bedrock chemically and physically as well as influence the style and pace of transport (Kelly et al., 1998; Gabet et al., 2003; Pawlik et al., 2016). Also, as mentioned above, trees limit the amount of water that flows to depth by taking up water in the rooting zone and transpiring it back to the atmosphere before it has a chance to interact with deeper material (Pavich et al., 1989; Moulton et al., 2000; Keller et al., 2006).

Together, these fundamental processes govern the physical evolution of hillslope form and lead to important fingerprints of biota on the terrestrial landscape (Dietrich and Perron, 2006). On human timescales, trees are often associated with landscape stabilization because dense root systems can create permeable material and bind it together in the root network (Prosser et al., 1995; Schmidt et al., 2001). These two effects of roots – creating permeability and binding weathered material – can discourage surface runoff and associated erosion and decrease the likelihood of downslope soil movement, including via landslides. Over time, however, the insertion of root and hyphae networks in soil and bedrock results in a significant amount of mechanical and chemical work that breaks, expands, and dissolves the near-surface material (Schaetzl et al., 1990; Van Scholl et al., 2008; Bonneville et al., 2009; Phillips, 2009). Therefore, although roots can stabilize soils, they can also act as preferential flowpaths for water that change the distribution of water pressure and sometimes promote landslides and erosion (Ghestem et al., 2011). Trees have thus been characterized as engines of weathering and erosion (Gabet and Mudd, 2010; Roering et al., 2010). It is unclear whether trees are more important as hillslope and soil stabilizers or as catalysts of bedrock erosion and soil formation globally (Brantley et al., 2012).

If one considers eroding, upland, soil-mantled landscapes underlain at depth by bedrock, material at depth must be moving up through the weathering zone over geomorphic timescales as material is removed near the earth's surface: this has sometimes been likened to a conveyor belt. We adopt a simple conceptualization of this weathering zone that differentiates fresh bedrock at depth from overlying weathered material. The uppermost layer of weathered material can move and is thus referred to as mobile soil (Table 1). Events such as landslides or tree throw can detach material from the immobile layer and move it rapidly into the mobile layer. These zones are depicted in Figure 2 wherein $h$ is the thickness of the mobile soil layer and $H$ is the thickness of the entire weathered zone – mobile and immobile – overlying bedrock. The relative values of $h$ and $H$ are thought to be set by the pace of erosion relative to the vigor and depth of biotic and abiotic weathering processes. In regimes lacking substantial deep weathering, the thicknesses of $h$ and $H$ may be effectively equivalent (Fig. 2a,c). In this case, trees can influence the conversion of subsurface material to mobile soil. By contrast, when $h << H$ (Fig. 2b,d), trees' direct influence on production of mobile soil is likely to be minimal. In these latter settings, weathered material may be sufficiently chemically depleted and mechanically weakened by the time it moves into the mobile soil layer that the contribution of tree root action is small compared to the sum total of reactions that produced weathered material at greater depths.

For $h \approx H$ regimes (Fig. 2a,c), the relationship among $h$, topography, and trees may depend on hillslope position (i.e., crest, sideslope, toe). For example, near ridge crests and in valley bottoms, the stress fields vary markedly, affecting the distribution of fractures (Wyrick and Borchers, 1981; St. Clair et al., 2015). An increase in the sharpness of a ridge (increased convexity) or an increase in topographic relief and narrow valley spacing can

generate stress concentrations sufficient to fracture bedrock along ridge crests and valley bottoms respectively (Miller and Dunne, 1996; St. Clair et al., 2015). Thus, topography affects fracture distribution, which in turn affects the efficiency of mobile soil production. These hypothesized interactions integrate processes that occur on highly variable timescales, making them challenging to model.

The aforementioned mechanistic interdependence of tree root activity and fractures emphasizes the role of tectonics in regulating CZ architecture. In landscapes where the ratio of the regional horizontal compressive tectonic stresses to near-surface gravitational stresses is relatively large, these stresses may promote the opening of fractures at great depth under ridges (St. Clair et al., 2015). One might expect that trees in such locations will have a limited role in shaping the CZ architecture because of the prevalence of deep regolith with deep or widely spaced fractures. By

contrast, in landscapes where the ratio of horizontal compressive tectonic stresses to near-surface gravitational stresses is relatively small, the opening of surface-parallel fractures in the near-surface might create a setting conducive to trees playing a critical role as near-surface opening-mode fractures are conducive to root growth. The roots can potentially extend fractures as well as detach and disaggregate bedrock, setting the thickness of the mobile soil layer ($h$) as formalized by empirical mobile soil production models (Heimsath et al., 1997). Such models

stipulate that subsurface material-root interactions (and thus mobile soil production rate) decrease with increasing thickness of mobile soil (Fig. 3). Numerous datasets of mobile soil production that use cosmogenic nuclides to quantify timescales support these concepts (Wilkinson and Humphreys, 2005; Heimsath et al., 2010).

The action of trees has frequently been implicated in controlling the dynamics of the mobile soil layer. For example,

researchers have suggested that trees can set i) the frequency with which soils are overturned and moved downslope by tree throw (Lutz and Griswold, 1939; Schaetzl et al., 1990; Schaetzl and Follmer, 1990; Norman et al., 1995); ii) the extent and magnitude of soil expansion through root network propagation (Brimhall et al., 1992; Hoffman and Anderson, 2014); and iii) the persistence of soil-stabilizing root networks (Denny and Goodlett, 1956; Schaetzl and Follmer, 1990; Norman et al., 1995). In most erosional settings, the depth of mobile soil, $h$, coincides with the

depth of physical or biological disturbance processes (Roering et al., 2010; Yoo et al., 2011). However, just because the depth of disturbances often correlates with mobile soil thickness, this does not necessarily demonstrate

causation. Furthermore, as alluded to in the last paragraph, roots are not limited to the mobile soil but are also commonly found in the weathered immobile layer, growing and taking up water (Graham et al., 2010).

### 1.4 Building and plumbing the critical zone

The implications of the ideas in these opening sections are explored in the hypotheses formulated below to explain the formation of the CZ and the movement of water within the CZ. Of particular interest are the widely held assumptions of each discipline that in some cases may be contradictory and may require more holistic understanding. While some of the hypotheses below may seem obviously true or obviously false to some practitioners in some disciplines, we argue that this just emphasizes the need for further research.

The hypotheses are separated into "building" and "plumbing" because it is clear that trees participate in both functions: trees build the critical zone by creating heterogeneity in the physical nature of weathered material, stabilizing this material, and plucking and mixing this material. But trees also plumb the critical zone by controlling the flow of water, exuding acids and organic compounds that solubilize material, and by hydraulically redistributing the water and solutes. However, we also recognize how difficult to impossible it is to separate these more physical, solid-phase and chemical, liquid-phase processes because, for example, the physical construct controls much of the water flow but the presence of water and solutes weakens the physical construct. We return to the interplay of building and plumbing at the end of the paper.

## 2 Hypotheses: How Trees Build the Critical Zone

### 2.1 Hypothesis 1: Tree roots can only physically penetrate and biogeochemically comminute the immobile substrate underlying mobile soil when that underlying substrate is fractured or pre-weathered.

Many authors have observed that roots can grow in close contact with weathered rock. However, few studies have systematically addressed lithological controls on root penetration into unweathered or weathered rock (e.g. Zwieniecki and Newton, 1994; Marshall and Roering, 2014). Although such close coupling has been used to argue that root growth can fracture rock, this may not be the case. Plant roots can exert axial pressure sufficient to create accommodation space as the roots lengthen in a soil matrix, but the material properties of soil, even a stiff clay, are vastly different from rock. Specifically, the fracture toughness, tensile or compressive strength of rock must be

overcome to lengthen or create fractures. Data summarized in the botany and agricultural literatures suggest that measured root pressures are unlikely to overcome the strength of all but the weakest bedrock: for example, laboratory experiments for peas indicate that the maximum measured axial and radial pressures of roots, 1.45 and 0.91 MPa respectively (e.g. Bennie, 1991), may only be large enough to break apart the weakest of sandstones. We therefore hypothesize, along with previous researchers (Zwieniecki and Newton, 1994), that tree roots can only grow into rock and promote weathering when fractures are already present or when the rock has already been weathered to some extent.

A large array of chemical and physical processes occur at the root-rock-regolith interface and some of these processes were recently reviewed, with an emphasis on the less direct (or obvious) process linkages (Pawlik et al., 2016). Although such processes have been studied to some extent, testing hypothesis 1 will require measuring root pressures for relevant species in natural settings in comparison to the strength of rocky material. Of course, laboratory experiments on root strength are poorly suited to real world bedrock settings both in terms of quantifying stresses over daily or annual time scales, and in replicating the fracture mechanics that result in actual root-fissure configurations (Gill and Bolt, 1955; Eavis et al., 1969; Misra et al., 1986; McCully, 1995; Gregory, 2006). Thus new techniques are needed to measure external root pressures *in situ*.

In addition to an incomplete understanding as to what controls the rates of root propagation into fractures or how the frequency of tree-driven processes may weaken rock, we also do not fully understand what controls the spatial distribution of roots within fractured material. Intriguingly, some research suggests that this spatial distribution may be influenced by mycorrhizal fungal communities (Egerton-Warburton et al., 2003). These communities may serve as frontier scouts for water and nutrients, especially in thick-rooted tree species with EMF as described in Section 1.1, and may complement roots in acquisition of these resources. Such exploitation could in turn generate stresses that might be sufficient to deform bedrock. If true, this implies that the microbial community may affect the manner and degree to which trees are able to convert material to soil. Of particular interest might be the possibility of phenomena such as stress corrosion cracking – chemical weakening of material that promotes fracturing. For example, we need to understand how chemical exudates near roots or fungal hyphae may be related to fracturing (Bonneville et al., 2009).

Of course, this endeavour to understand root-generated fracturing strongly depends on our understanding of the mechanical properties of the material to be fractured. Under mobile soils that are thin, the patterns of rock fracturing

and weathering may be an important limit on the rate of detachment of sub-soil material, and on the size of detached fragments incorporated up into the mobile soil. In such cases, trees affect the efficiency of mobile soil production (Jackson and Sheldon, 1949; Marshall and Roering, 2014). This contrasts with settings with thick regolith (Chadwick et al., 2013), whereby climate or slow erosion rates diminish the role of trees in the production of mobile

soil thickness to the point that roots do not penetrate deeper than $h$ (see Table 1 and Figure 2). The fracturing of bedrock has been well studied in structural geology and geomechanics. While the substantial literature generated by those fields is highly useful, the partially weathered status of immobile material in the CZ likely has a profound influence on mechanical properties, and we know less about the physical attributes of these weathered materials. This points toward the need for a systematic and comprehensive analysis of rock properties as a function of

weathering state (Selby, 1993; Murphy et al., 2016).

**2.2 Hypothesis 2: In settings where the thickness of weathered material, *H*, is large, trees primarily shape the CZ through biogeochemical reactions within the rooting zone.**

The mobile soil layer contains the highest densities of roots and mycorrhizal fungal communities. According to hypothesis 1, tree roots can penetrate material underlying the mobile soil when this underlying substrate is fractured

or pre-weathered and $h \approx H$.  However, if the total layer of weathered material ($H$) is very thick, tree roots do not commonly reach unweathered bedrock. In regions where $h << H$ (Fig. 2B), therefore, we hypothesize that the most important role that living trees play in formation of mobile soil is not related to insertion of roots into bedrock fractures. Rather, the major effect is more likely biogeochemical in nature and limited to upper layers.

Of particular interest with respect to this hypothesis is soil associated with the rhizosphere (Hiltner, 1904; Hartmann et al., 2008). The rhizosphere is the most biologically and chemically active frontier of the soil (McNear, 2013) because this is where compounds are released which directly and indirectly affect soil minerals (Philippot et al., 2013). Specifically, roots provide carbon for the microbial and fungal communities (Berner et al., 2003; Calvaruso et al., 2009; Calvaruso et al., 2014; McGahan et al., 2014). In return, mycorrhizal fungi and associated bacteria

generally increase the availability of nutrients to the trees (e.g. van Scholl et al., 2006a; van Scholl et al., 2006b; Balogh-Brunstad et al., 2008a; Calvaruso et al., 2009; Bonneville et al., 2011; Smits et al., 2012; Ahmed and Holmstrom, 2015).

Two direct pathways by which nutrients are extracted from soil minerals are i) dissolution driven by protons

released into the rhizosphere in exchange for other cations; and ii) chelation with organic compounds released into

the rhizosphere by fungi (Leake et al., 2008; Smith and Read, 2008; Ahmed and Holmstrom, 2015; Finzi et al., 2015). Other more indirect pathways also are hypothesized to be important, including exudation of reductive compounds (Fimmen et al., 2007), pumping of water up and down (Fig. 2d) within the soil to access different minerals (Dijkstra and Smits, 2002), effects on temperature and water throughput (Moulton et al., 2000; Keller et al., 2006), and the increase in chemical affinity that results from uptake and sequestration of reaction products.

In addition, plants can also indirectly promote weathering by secreting bio-signaling molecules to activate their mycorrhizal networks and associated micro-organisms (Deveau et al., 2012; Venkateshwaran et al., 2013). Such secretions initiate a cascade of reactions that then allows them to take up weathering products. Ectomycorrhizal fungi also are able to actively decompose organic matter to acquire nitrogen and phosphorus (Marschner, 2011; Reed et al., 2011). In fact, at the watershed scale, many studies have shown that trees can increase mineral dissolution rates (Berner et al., 2003; Calvaruso et al., 2009; Calvaruso et al., 2014; Augustin et al., 2015) compared to rates observed for rock areas that are bare or lichen- or moss-covered.

A big unknown in regard to the chemical effects of biota is the mycorrhizal fungal community (Grantham et al., 1997; Balogh-Brunstad et al., 2008b; Graham et al., 2010). Numerous experimental studies have shown that roots and their symbiotic fungi constantly forage and biosense nutrient sources (Leake et al., 2008; McNear, 2013), and thus roots and fungi perhaps access nutrients down to several meters in depth (Graham et al., 2010; Hasenmueller et al., 2017). However, studies of such fungi below the mobile soil are limited. Where hyphae penetrate downward, there is a large potential for mycorrhizal fungi to weather the immobile substrate at depth. Since roots are sometimes observed to penetrate the immobile weathered material even in humid forested regions (Hasenmueller et al., 2017), mycorrhizal fungi undoubtedly also explore this zone and contact immobile material (Rosling et al., 2003; Graham et al., 2010; Callesen et al., 2016). To understand such phenomena will require better techniques to map fungal presence or absence and further exploration of how and when secondary phases such as clays, organo-amorphous phases and oxides seal the surfaces of minerals from dissolution (Kleber et al., 2007; Zhu et al., 2014). The fungal contribution – and more broadly, the soil microbial contribution – to weathering remains a largely unexplored frontier in CZ science. We need to collect deep cores into weathered material and save the material not only for physical and chemical analyses but also for biological, molecular analyses and DNA sequencing, with particular emphasis on roots and fungi. Understanding the large data sets that can result from these efforts will also require new capabilities in data analysis.

## 2.3  Hypothesis 3: In forested uplands, the thickness of mobile soil, *h*, can evolve toward a steady state because of feedbacks related to root disruption and tree throw.

Geomorphic and geochemical process models imply critical zone properties will tend toward a time-independent thickness of mobile soil, *h*, if tectonic forcing (e.g. uplift rate) and climate forcings (e.g. rainfall, temperature, and seasonality) are constant.  In this hypothesis, we posit that the thickness, *h,* of the mobile soil under a forest is maintained mainly by soil churning and disturbance of the underlying immobile substrate via root-wedging and tree-throw. We also implicitly argue based on the previous two hypotheses that such a steady state is only likely for the endmember case when $h \approx H$ (Fig. 2a)*.* Under these conditions, we hypothesize that trees act as the main feedback that maintains a steady state value of *h* by coupling erosion with weathering (Figure 2c).  Steady state is most likely when tectonic or topographic stresses promote near-surface fracturing and weathering (see hypothesis 1) and when transport processes are sufficiently fast such that erosion is not rate-limiting. Instead, this steady state is likely when detachment of mobile material from underlying material limits the rate of overall loss of material from the system (i.e., denudation).

In such detachment-limited settings, the ability of tree root networks to disturb shallow weathered immobile material likely depends on the material properties of that material. In other words, when $h \approx H$, trees have access to the immobile weathered substrate at depths greater than *h* if this material is fragmented or weathered and in this case this material can be uplifted by roots (Figure 2A).  These processes may affect whether the mobile soil production rate exhibits a humped relationship such that it increases and then decreases with mobile soil thickness as exemplified in Figure 3 (Cox, 1980; Furbish and Fagherazzi, 2001). For example, empirical data (Heimsath et al., 2001; Gabet and Mudd, 2010) from the heavily forested Oregon Coast Range are generally consistent with the humped model predictions of increasing and then decreasing mobile soil production rate with increasing mobile soil thickness. However, an exponential soil production function may equally well fit the data (e.g. Heimsath et al., 2005). In that case, root density and thus thickness of material disturbed by tree throw might depend on factors such as rock strength or fracture density as well.

The nature of the feedbacks that explain how a steady state thickness might develop (or even whether a steady state thickness ever occurs) are not well understood. Numerical simulations have been used in the geological literature to explore tree-driven mobile soil production: these models are consistent with a 'humped' mobile soil production function (Fig. 3). Such a function predicts maximum production rates at values of mobile soil thickness that are non-zero (Gabet and Mudd, 2010).  This leads to the idea that a complex relationship likely exists between mobile

soil thickness and tree density. One explanation for this functional relationship emerges from the *a priori* stipulation that tree density increases with mobile soil thickness. As mobile soils become sufficiently thick, however, Gabet and Mudd (2010) have argued that a negative feedback must exist. Specifically, as $h$ increases, tree density continues to increase but the frequency of immobile material-root interaction decreases, resulting in a reduction in the rate of mobile soil production. In fact, however, in landscapes with maturing forests and where mobile soils are not extremely thin or very infertile, tree density becomes independent of mobile soil thickness because tree density becomes dictated mostly by canopy closure and differential mortality of smaller, light-limited individuals ("self-thinning" (Lonsdale, 1990)). Thus, as forests mature, tree density is affected more by competition among trees of different age and size than by mobile soil thickness. The negative feedback that slows down mobile soil production (Fig. 3) as mobile soil thickness increases must therefore be related to phenomena other than tree density. Some have argued, for example, that porewater chemistry might provide a negative feedback such that thicker weathered material produces less corrosive fluids at depth that could slow down the rate of production of weathered material from unweathered material (Fletcher et al., 2006). Finally, the idea of trees acting as feedback mechanisms controlling mobile soil thickness is predicated upon the assumption that the subsurface material is amenable to disruption by tree roots – and this may not be the case in the absence of fractures and weathering in the underlying immobile material as discussed in hypothesis 1.

A corollary to this hypothesis and hypothesis 2 is that trees can contribute chemically to altering minerals when $h << H$, but cannot physically or chemically set the rate of formation of mobile soil from underlying material when $h << H$ because the subsurface injection of carbon at depth is minimal. When $h << H$, solute fluxes, transmissivity, grain size distribution and other near-surface attributes of the mobile layer may vary significantly with time and therefore may not reach a steady state. If a steady state is reached under these conditions, other attributes of erosion and weathering unrelated to trees presumably maintain the stable value of $h$.

In the two end-member cases of $h \approx H$ and $h << H$ (Fig. 2), roots and rhizospheric microbiota may function in two different ways. When $h \approx H$ (Figure 2Aa), roots and associated microbial communities interact significantly with both the mobile soil and the upper layers of unweathered bedrock, actively weathering primary minerals containing many macronutrients (e.g. P, K, Mg, Fe, and Ca). Uptake of these nutrients into hyphae and roots nourish the plants. In fact, if P is present at a low concentration, some root-associated fungi can "biosense" P hotspots and proliferate into those locations (Leake et al., 2004). This has not been shown for other elements (Wallander and Ekblad, 2015) although upward pumping of elements such as Ca has been hypothesized (Dijkstra and Smits, 2002). We expect

that water availability in the soil most likely influences all these processes that are mediated by mycorrhizal fungi (see Fig. 4 and hypothesis 4).

In contrast, when $h << H$, roots and associated mycorrhizal fungi have little to no contact with the unweathered bedrock (Figure 2b). In this case, roots and associated micro-organisms are not likely to access nutrients in the bedrock itself and therefore must recycle nutrients (Fig. 2D) by decomposing organic matter and capturing nutrients from water infiltrating downward in the profile of mobile soil and immobile weathered material (Smith and Read, 2008; Marschner, 2011). In addition, the degree to which tree species rely on their mycorrhizal fungi depends on the thickness of their roots and the type of mycorrhizal fungi (Brundrett, 2002; Chen et al., 2016; Cheng et al., 2016). Roots and associated microbiota may be able to shift between actively weathering primary mineral phases to purely recycling nutrients from organic matter and soil depending upon the relative magnitude of $h$ with respect to $H$ in different climatic, lithologic, and tectonic settings.

In steep forested hillslopes, trees may impart a distinctive topographic signature that results from these process interactions. For example, analysis of airborne lidar for western Oregon hillslopes (35-40º) shows that pit-mound features generated by tree turnover dominate landscape morphology at length scales less than 8 m while hillslope-valley landforms characterize landscape form at longer length scales, and these features are observed at hilltops and hillsides regardless of slope (Roering et al., 2010). Ground-penetrating radar reveals a similar topographic pattern along the interface between weathered mobile and immobile material, which results in highly variable mobile soil thickness (Heimsath et al., 2001). On these closed-canopy coniferous slopes with typical mobile soil thickness values of 0.5 to 1.0 m, large roots (>10 cm diameter) are observed to utilize shallow fractures in rock to reach depths of 2-3 m immediately below tree stems. In these below-stem zones, root penetration is observed to be accompanied by disaggregation of material. Although at any given time the basal area of stems only occupies <5% of the forest floor, the regional average erosion rate (~0.1 mm/yr) and recurrence interval of stand-resetting fires (250 to 400 yrs) imply that virtually all parcels of immobile weathered material and mobile soil are impacted by below-stem large root penetration during their exhumation to the land surface. In other words, when erosion rates are not overly fast, tree roots interact with or 'touch' the vast majority of shallow immobile weathered material (as well as mobile soil) that eventually erodes from the hillslope and is delivered to stream networks (Roering et al., 2010). Some have inferred from this that trees influence not just their near-surface terrestrial environment but likely contribute to the grain size distribution that participates in nearby stream incision or that supports nearby aquatic ecosystems (Sklar, 2017).

In contrast, in a relatively moist, mixed temperate, closed-canopy forest in a Pennsylvania catchment developed on grey shale with somewhat more gentle slopes and erosion rates of approximately 0.03 mm/y (West et al., 2013), only relatively fine roots (e.g < 5 mm) are observed penetrating deeper than 1 meter into the immobile weathered material (Hasenmueller et al., 2017). The fine roots are typically observed when this rocky immobile material breaks apart into fractures where the roots have penetrated (Hasenmueller et al., 2017). This location also exhibits pits and mounds that define the topography at tens of meters length scales, hillslope-valley landforms at longer length scales, and mobile soil that varies in thickness from tens of centimeters at ridgetops to approximately a few meters in valley bottoms and swales. The lack of a high density of roots at depth is not because of a lack of fractures in the shale because the upper 5 to 8 m of the rock is highly fractured, a characteristic attributed to the periglacial climate during the Last Glacial maximum (Jin et al., 2010). Although deep fine roots are observed, their density is very low compared to the roots in the upper 30 cm of soil where the trees get most of their water (Gaines et al., 2016). In other locations, rooting depth is not only controlled by the availability of fractures in the rock, but also by the demand for deeper sources of water (Schenk 2008). In the humid, shale catchment in Pennsylvania, this demand for water is not high for most of the year because frequent showers during the summer wet the surface soil layers and transpiration is tempered by relatively low winds, high humidity and modest temperatures. Rooting depth may thus be considerably shallower in more mesic environments than in more arid environments.

Clearly, the systematic feedbacks between roots and rocks remain to be investigated within this concept of steady state thickness of mobile soils. The research agenda here is wide open. Open questions abound. How long does it take to achieve steady state and how do these timescales compare to the frequency of significant perturbations? What are the implications of our two end-member scenarios ($h{\sim}H$ and $h{<<}H$) in terms of how trees plumb the critical zone (see Section 3)? How do disturbances on the hillslope to landscape scale affect the role of trees in building, maintaining, and plumbing the critical zone? How can this framework of trees creating and maintaining their CZ resources be extended to depositional settings, glaciated landscapes, etc.? Furthermore, how does the ecological functioning of trees differ, including their access to nutrient resources such as phosphorus, under the global range of conditions? Answers to such questions will largely come from careful studies of mobile soil thickness and its relationship with tree root distribution as a function of tectonic, lithologic, and climate conditions in different settings, and then careful comparisons and modelling efforts to explain differences and similarities.

**2.4 Hypothesis 4: In settings where $h << H$ and the rate of uplift and erosion are low, the uptake of phosphorus into trees is buffered by the fine-grained fraction of the soil, and the ultimate source of this phosphorus is dust.**

Since phosphorus (P) is a rock-derived nutrient, its availability to an ecosystem is usually controlled by the concentration and reactivity of the phosphorus-containing mineral apatite in the rock (Boyle et al., 2013).

Furthermore, the amount of mobile and readily available P in soil is usually low because P is easily taken up by organisms or sorbed onto mineral surfaces. Given these attributes, clay, organic matter, and iron oxide surfaces comprise a colloidal "plasma" within soil that can buffer P concentrations. The plasma provides different types of sorption sites that can hold P either strongly or weakly depending on their chemical character (Hemwall, 1957). On relatively long time scales, P availability is also affected by the rate that the unweathered rock containing apatite is

advected upward into the weathering zone by uplift and erosion (Porder et al., 2007; Vitousek et al., 2010). P can also be added to the surface as finely divided mineral aerosol that can weather to release P relatively quickly.

Some rocks are naturally low in P, and ecosystems growing on such rocks must strongly recycle P or be replenished by inputs of mineral dust. However, even for lithologies with abundant P, the main source of this macronutrient

can still be dust (Okin et al., 2004; Porder and Chadwick, 2009; Aciego et al., 2017) depending on the rates of uplift and weathering. Specifically, slow rates of uplift and erosion lead to long mineral residence times within the weathering zone (c.f. Porder et al., 2007) and loss of P by leaching. Addition of mineral aerosols at the surface provides a rapidly available source of P, both because of its fine grain size and because it is deposited into the most acidic, organic-rich part of the profile. The importance of dust inputs of P to ecosystems has been observed in arid

as well as humid tropical systems (Chadwick et al., 1999; Pett-Ridge, 2009). We hypothesize that dust will be the predominant source of P in systems where the thickness of the mobile soil $(h) <<$ thickness of the total mobile soil + immobile subsoil substrate $(H)$.

The weathering products derived from different rocks also have a strong control on the availability of P to trees.

As part of this hypothesis, we posit that for rocks such as basalt and shale that produce soils with high plasma : skeleton ratios (e.g. a large fraction of the soil is composed of secondary clays and colloids as opposed to sand or pebbles), the proximal control on P availability lies in the plasma surface area. By contrast, for rocks that produce low plasma : skeleton ratios such as granite and quartz-rich sandstone, we expect that uplift (erosion) will impose an absolute constraint on P availability that is far less buffered by proximal controls such as plasma sorption. Those

lithologies that form soils with low plasma : skeleton ratios are more likely to have P-limited ecosystems (Hahm et al., 2014) and therefore be influenced by differences in dust inputs (Aciego et al., 2017). Such low plasma : skeleton

lithologies also are more likely to develop strong local P gradients due to hydrological redistribution along hillslopes (Khomo et al., 2013; Bern et al., 2015). This can in turn create local patchiness in vegetation type and productivity (Venter et al., 2003).

To understand sources and fates of P in forest ecosystems, researchers need to evaluate the balance among processes affecting both the absolute amount of P and the rate at which it becomes available to trees. They must find ways to identify dust in soils, including fingerprinting by mineralogical, size, trace element, and particle morphological analysis. They need to quantify uplift (or erosion) rates and to understand how erosion may respond to short-term perturbations such as logging. They need to document plasma : skeleton ratios as a way to index the sorptive

capacity of the soil and to determine the point when P sorption capacity has been reached. A starting point for this work might be to identify ecosystems within the same climate zone that survive on rocks that weather to differing amounts of plasma and skeleton under different uplift rates but with similar dust inputs. At the other end of the spectrum, ecosystem and weathering models can be coupled to evaluate plausible rates of release and sorption of P depending on differing suites of starting minerals.  All such approaches could be used to explore the role of dust

and plasma in P availability in soils.

### 3 Hypotheses:  How Trees Plumb the Critical Zone

### 3.1 Hypothesis 5: In settings of limited water availability, trees maintain the highest density of functional roots at depths where water can be extracted over most of the growing season with the least amount of energy expenditure.

Water potential is defined as the potential energy per unit volume of water within a soil-plant system relative to pure water at sea level (Kramer and Boyer, 1995). Generally, water in the soil-plant system is at a negative potential, i.e., the plant is "sucking" water out of its environment under tension. Water potential is affected by the gravitational, turgor, osmotic, and matric potential of water in the system (Kramer and Boyer, 1995).  Briefly, these terms refer to the hydrostatic head, the pressure associated with cell expansion in growing tissues, the tension

related to the solute content of the water in different reservoirs, and the surface tension that arises between water and solids.

A water molecule will move to the root if the water potential in the soil is higher than the water potential in the root.  Of particular importance for plants is the matric potential of soil water. At some times of year or in some

environments, the matric potential can be more negative than the lowest potential from which plants can access appreciable water, i.e., the so-called wilting point (Fig. 4). However, this concept may be inappropriate for trees because it is based on the concept of a "standard [herbaceous] crop plant". Within the soil matrix, a plot of matric potential versus the volume of water can be conceptualized as delineating different water reservoirs ranging from

water that drains freely due to gravity to so-called hygroscopic water which may not be accessible directly to roots except under certain conditions (Fig. 4). Field capacity is operationally defined as the water potential associated with the moisture remaining after a soil has been fully wetted but any excess water has drained away. Between the wilting point and field capacity is the potential of capillary water: this water is held by surface tension the soil matrix and is readily accessible by plants.

If water in the upper 10 cm of soil is of equal water potential to that at 1 m depth, then trees will use the surface water first, both because it requires less energy to move the water to the leaves and because there is typically much greater root length near the soil surface (Green and Clothier, 1999). Higher root length density means that the distance from bulk soil to root is shorter, on average, and this shorter distance of transport enables the plant to take

water up quicker. However, if soil water potentials are low (more negative) in the surface layers but high at depth, some trees may instead acquire a substantial portion of water at depth instead of from the surface (Jackson et al., 1999).

Some studies have identified circumstances where despite groundwater being readily available within 0.5 m of the

surface, tree species instead use rainfall at shallower depths (Busch et al., 1992; Snyder and Williams, 2000). For example, after one rainfall event, as much as 40 – 50% of tree sap water in one system was shown to be derived from rain water (White et al., 1985). Such opportunistic use of water is a strategy consistent with the expectation that new, shallow sources of water from a rainfall event are energetically less costly to obtain because they are present at a higher water potential and are present in the zone of greater root length density. We also know that

nutrients that plants require are generally present at higher concentrations in surface soils because they are taken up into plants and then returned to the land surface through leaf litter or other decaying plant material (Jobbagy and Jackson, 2004). Strategically, many trees take up shallow water instead of deeper groundwater at least partly because the root length density is generally lower at depth.

Similarly, trees growing alongside perennial streams in arid regions do not necessarily use what seems to be the most easily accessible stream water. Instead trees may access soil water from either deeper layers (Dawson and

Ehleringer, 1991) or from deeper saturated soils where a high fraction of roots reside (Bowling et al., 2017). In those locations, it is possible that the root density is larger at depth than near the surface, allowing water to be taken up from depth even during the parts of the year when plentiful water is available in the stream. This idea has led to the view that plants may utilize different niches (Silvertown et al., 2015) by partitioning their roots according to the hydrological conditions of different layers (e.g., *Walter's Two-Layer Hypothesis*). Specifically, Walter's hypothesis states, in part, that shallow and deeply rooted plants do not compete for the same water resources (Walter, 1939; Ehleringer et al., 1991; Weltzin and McPherson, 1997; Schenk and Jackson, 2002a; Schwinning, 2008; Holdo, 2013; Ward et al., 2013).

From these observations emerges our hypothesis, namely that trees grow high root densities at depths where water is most easily extracted for the largest portion of the growing season. Thus, during time periods of the year where water is available at depths that generally do not have water, trees will continue to extract water from other depths where they have more dependably found readily available water. A corollary to this hypothesis is that the root length density is a map of where water is most likely to be present for much of the growing season when trees need water. Such corollaries can be tested by measurement of root length densities and water usage by trees in soils in different landscape positions, on different lithologies, and on soils developed in different climates.

### 3.2 Hypothesis 6: Trees grow the majority of their roots in the zone where the most growth-limiting resource is abundant, but they also grow roots at other depths to forage for other resources and to hydraulically redistribute those resources to depths where they can be taken up more efficiently.

This hypothesis is a corollary of hypothesis 5 where we hypothesized that the depth where trees in water-limited environments grow roots is intimately linked to where they are able to acquire water while conserving the most energy over most of the growing season. However, uptake of water and nutrients need not be tightly coupled (Pate et al., 1998). While some plant species rely mainly on deep soil moisture for transpiration (Kurc and Small, 2007; Kurc and Benton, 2010; Cavanaugh et al., 2011), their nutrient uptake may be uncoupled from this water uptake if the nutrients are only present in shallow soil or near decomposing leaf litter. On the other hand, significant pools of some nutrients may be found in deeper soil layers closer to unweathered bedrock (McCulley et al., 2004; Maeght et al., 2013). Such deep nutrient access might provide an explanation for observations of some low-density root growth in deep fractured rock or soil even when most of the roots grow in the shallow, wetter layers (e.g. Hasenmueller et al., 2017). In fact, some trees in more arid environments have so-called "dimorphic root systems".

These trees produce abundant fine roots in the surficial soil to recover nutrients from fallen leaves, and they grow abundant deep roots with highly efficient transport anatomies to acquire sufficient water from deeper reservoirs (Pate et al., 1998).

An important aspect of this hypothesis is the phenomenon of hydraulic redistribution. Such redistribution may provide another mechanism for plants to solve the problem of different spatial distributions for water versus nutrients (Caldwell et al., 1998) and could be important in keeping fine roots alive in arid systems by reducing loss to evapotranspiration (Burgess et al., 1998).  Hydraulic redistribution is the process by which plants redistribute water in the soil profile from moist to dry regions using their root systems (e.g. Caldwell et al., 1998; Oliveira et

al., 2005). Specifically, hydraulic redistribution can bring water (and perhaps nutrients) in some soils from depth to the dry surface, so that at night, the rhizosphere is moistened, allowing for nutrient solubilization as well as decomposition of organic matter (Armas et al., 2012).  Although not proven, Ca redistribution from deep to shallow has been hypothesized in at least one soil system (Dijkstra and Smits, 2002). Some argue that trees move water around in the soil to protect and retain nutrients (Burgess et al., 1998).

To explore this hypothesis will require careful studies that determine the spatial and temporal distribution of root length density, water isotopes, nutrient distributions and fluxes, and hydraulic redistribution. For example, some stable isotope studies (e.g. Phillips and Ehleringer, 1995) and sap flow measurements linked with soil moisture measurements at depth (e.g. Cavanaugh et al., 2011) have identified cases in which plants with roots mostly near

the surface still rely mainly on deep soil moisture for transpiration. For those systems, we infer that the shallow roots are grown densely to provide growth-limiting nutrients; however, such an inference should be tested.  Similar studies have also identified cases in which at least grasses have grown a high density of roots at depth and actually seem to prefer taking water up from shallow reservoirs (e.g. Nippert and Knapp, 2007). For those cases, plants may be growing deep roots as a competitive strategy to limit uptake of water and nutrients by neighbors (McNickle and

Dybzinski, 2013). One way to investigate this hypothesis and hypothesis 5 is to explore root distributions in the context of mineralogy, bulk chemistry, plasma and skeleton content, and water distribution.

### 3.3 Hypothesis 7: Trees rely on matrix water in the unsaturated zone that at times may have an isotopic composition distinct from the gravity-drained water that transits from the hillslope to groundwater and streamflow.

Given the importance of tree roots in affecting soil permeability, trees play a significant role in routing water within the critical zone. Specifically, water can pass through soil matrix as infiltration or it can bypass much of the bulk soil and flow through macropores, the majority of which are thought to be related to roots. Specifically, root-related macropores can contain live roots, dead roots, or dead and live roots together (Ghestem et al., 2011).

Ecohydrological separation – defined as trees using water of a character different from the gravity-drained water found in soils, in saprolite or in groundwater and streams – has been hypothesized to be common based on a recent meta-analysis of isotope ecology literature (Evaristo et al., 2015) and global remote-sensing data based on the deuterium composition of atmospheric vapor (Good et al., 2015). These and related studies (e.g., Brooks et al. 2010) suggest that trees rely on water present in the unsaturated zone and this water may have an isotopic composition distinct from the gravity-drained water that transits the hillslope to become groundwater recharge and streamflow.

This "two-water-world" hypothesis (McDonnell, 2014) is controversial (Berry et al., 2017; Sprenger et al., 2016) and could be at odds with the existence of subsurface reservoirs such as layers of saprolite and fractured, partly weathered immobile material that hold water that is accessed by trees (Oshun et al., 2016). For example, in seasonally dry climates, trees may derive a significant portion of their moisture from immobile weathered material well below the soil (Zwieniecki and Newton, 1996; Graham et al., 2010; Nie et al., 2012). In arid or hyperarid systems, the fraction of use of deep water increases as annual rainfall decreases (Dawson and Pate, 1996; Dawson et al., 2002). Such deep water resources link deep unsaturated zone moisture to the atmosphere and hydrologic cycle through root uptake and transpiration. Yet, the evidence for ecohydrological separation suggests that trees may not always use gravity-drained water if other, more energetically available sources are present.

The evidence for ecohydrological separation (McDonnell, 2014; Evaristo et al., 2015; Good et al., 2015) suggests that plants are sometimes using water from unknown depths and that the water potentials are different from what might be considered the "crop plant" wilting point, e.g., < -1.5 MPa (Kramer and Boyer, 1995). Furthermore, in some cases, Evaristo et al. showed that gravity-drained and transpired waters were not isotopically distinct. These observations document that our understanding of how water is obtained by roots in the deeper subsurface is lacking.

Some of the paucity of knowledge is related to questions of physiology and some to subsurface structure and character (Washburn and Smth, 1934; Walker and Richardson, 1991; Hiscock et al., 2011).

Methods to extract and measure tree water sources are currently being refined and improved to test hypothesis 7. Currently, the techniques for sampling soils or plants can yield waters with different isotopic signatures and it is not known if these differences are caused by the extraction methodology or differences in the water samples themselves. There have been a number of recent papers building upon the early work in Graham Allisons' laboratory exploring water isotope fractionation in subsurface pools (Allison et al., 1983). This new work investigates methodologies of extraction, isotope fractionation during water uptake by plants, and interpretation of isotope data (Oerter et al., 2014; Orlowski et al., 2016a; Orlowski et al., 2016b; Oshun et al., 2016; Zhao et al., 2016; Gaj et al., 2017; Johnson et al., 2017; Vargas et al., 2017). These papers provide new insights at the same time that they add to the ongoing controversy about what explains water isotope variation in the many possible subsurface pools, highlighting the need for research. Nonetheless, an additional intriguing observation is that many trees with mycorrhizal fungal associations appear to have a mechanism for tapping water below the agronomically-defined soil wilting point of cultivated plants (also see hypothesis 8). This should not surprise us since we have known that the wilting point of a crop plant and a tree are rarely, if ever, the same: tree values can be much, much lower (Martinez-Vilalta et al., 2014; Meinzer et al., 2016). So the "two-water world" hypothesis must now be thoroughly tested in the context of water potential measurements and theory (see hypothesis 5 and Bowling et al. (2016)) for how plants are known to take up water. Research is also needed to investigate the physical and chemical effects on the isotope composition of water in the subsurface (Oshun et al. 2016) and on new observations about fungal access to water as described in hypothesis 8.

### 3.4 Hypothesis 8: Mycorrhizal fungi can use matrix water directly but trees can only use this water by accessing it indirectly through the fungi.

Mycorrhizal fungi may play an important role in water acquisition (Duddridge et al., 1980; Augé, 2001; Plamboeck et al., 2007; Bárzana et al., 2012). Hyphae, fungal threads emanating from the root, may allow a plant to access water from water-filled pores that are too small for the roots. Arbuscular mycorrhizal (AM) fungi, for example, have hyphae with diameters between 2-20 μm, which is typically an order of magnitude or more smaller than roots. Hyphal length density can vary between 1 and >100 m per gram of soil (Smith et al., 2010). Thus, mycorrhizal hyphae may access water not available to plant roots, presumably because fungal hyphae can penetrate small water-filled pores to a greater extent than the larger roots (Bornyasz et al., 2005; Allen, 2007; Graham et al., 2010; Lehto

and Zwiazek, 2011). Thus, mycorrhizae may be a factor that facilitates plant access to rock moisture and matrix waters that would otherwise be inaccessible to roots. Although water in the rock matrix may not actually be held at tensions higher than the permanent wilting point, the pore network may be so small that only hyphae can penetrate. These hyphae-pore interactions also have the potential to affect *h/H* through mineral plucking, and

changes in pH or redox status (see hypothesis 2).

Although it makes physical sense that hyphae may penetrate pores in rock matrix that are smaller than roots can penetrate, many researchers are not convinced that mycorrhizal fungi play an important role in acquiring water at water potentials beyond the wilting point (Kothari et al., 1990; George et al., 1992; Koide, 1993; Bryla and

Duniway, 1997). For example, one counterargument is that the hyphae have high axial resistance to water flow because of their small diameters and their lack of vessel-like structures: this observation might lead one to argue that flow rates in hyphae simply are too slow to appreciably contribute to transpiration directly (Koide, 1993). Most improvements in plant growth or survival related to mycorrhizal fungi are considered to result not from water uptake but rather from the indirect effects of fungal-mediated nutrient acquisition and improved plant nutrition

(Kothari et al., 1990; Bryla and Duniway, 1997). In this regard, EM and AM fungi may differ significantly. Unlike AM, EM fungi are capable of forming relatively large-diameter rhizomorphs made of fused hyphae where hydraulic conductance is high enough to contribute significant water to plants (Brownlee et al., 1983; Warren et al., 2008). Of course, these larger hyphae may be unable to access the finest matrix pores.

Clearly, to explore hypothesis 8 requires not only assessing the size and distribution of small pores in unweathered rock, immobile weathered material, and soil (Bazilevskaya et al., 2015), but also which pores allow hyphal access and water and nutrient uptake (Graham et al., 2010). Mapping of fungal hyphae in mobile soil, immobile weathered material, and unweathered rock will be required. Techniques might utilize observations in pit walls or impregnated blocks or excavations. Tracer studies that could assess different types of water inside different regolith types or

inside fungal hyphae would also be of interest.

### 3.5 Hypothesis 9: Even trees growing well above the valley floor of a catchment can directly affect stream chemistry where changes in permeability near the rooting zone promote intermittent zones of water saturation and downslope flow of water to the stream.

One of the outstanding research questions concerning small catchments is how to predict the relationship of solute

chemistry and discharge as a function of variations in precipitation (Godsey et al., 2009). In catchments, many of

the nutrients and other solutes added to rain water as it transits through hillslopes to the bounding streams are added from weathering reactions in the soil. These reactions are more likely to occur in the matrix, where the surface area wetted by porewater is high. However, as discussed in hypothesis 7, pore water in the matrix does not generally drain by gravity. In fact, pore waters in gravity-draining pores in regolith may mix with matrix pore waters only under water-saturated conditions. Under these conditions, nutrients and other solutes in matrix waters mix with the gravity-drained waters and then move to the stream. Therefore, the matrix will only deliver water to the stream if the hillslopes are hydrologically connected to the stream.

Given these observations, it is difficult to imagine how trees growing high on hillslopes might affect stream chemistry (Fig. 5). For example, hillslopes are mostly disconnected from streams during baseflow, and stream chemistry is not likely to be strongly influenced by trees during those time periods. In contrast, during hydrologically connected periods, we hypothesize that trees on hillslopes can impact stream chemistry detectably. Predicting the impact of trees on stream chemistry therefore depends on understanding the degree of connection between the hillslope and the stream (Herndon et al., 2015). According to this hypothesis, biogeochemical processes such as cation exchange occurring in matrix waters can influence ecological responses in streams under conditions of high connectivity (e.g. Green et al., 2013).

Hydrologic connectivity can be quantified in multiple ways (Larsen et al., 2012; Spence and Phillips, 2015). However, metrics of connectivity that work well in some settings are not always transferable to different locales (James and Roulet, 2007). We hypothesize that changes in connectivity are dictated by the extent of water saturation and the nature of the architecture of the critical zone in any given catchment. For example, we assume that there is usually a sharp decrease in vertical hydraulic conductivity at the base of the mobile soil layer (Fig. 2). At this interface, water may pond and create a transiently saturated layer that can drain via macropores laterally and vertically, allowing matrix waters to preferentially mix along the mobile soil - immobile material contact. If the perched water zone connects all the way down the hillslope, water can flow downslope to the stream. A hypothetical geometry for this is shown for the connected gravity-drained water in Figure 5. Spatial heterogeneity in the contact between the mobile and immobile layers will greatly influence the subsurface drainage to the stream. Specifically, such subsurface topography in many locations is characterized by depressions that "fill and spill" depending upon the extent of saturation (Tromp-van Meerveld and McDonnell, 2006).

Based on hypothesis 1, it is possible that the location and depth of the depressions at the base of the mobile layer that "fill and spill" and control hillslope-stream connectivity are related to the penetration of tree roots into the layer of weathered immobile material and the effects of tree throw (Fig. 2). Such penetrating roots (see hypothesis 1 and the discussion for hypothesis 3) can have a strong influence by plucking rock material and creating the rough undulations at the interface between the overlying permeable layer and the underlying more impermeable layer (Fig. 2). Rooting depths in systems where $h \approx H$ may even be deep enough to interact with the bedrock as well as the immobile weathered material, and can draw up water from below (hypothesis 2) as well as enhance physical and chemical weathering (hypothesis 1). Furthermore, fracture density and development both affect the tension under which water is held in rock and soil, potentially affecting timescales of movement of water and solutes, as well as chemical weathering. All of these likely comprise feedbacks that affect the spatial pattern of roots and mycorrhizal hyphae at various depths and create a subsurface mosaic of hydrological connectivity. In fact, some researchers have mapped lateral subsurface water flow and attributed it entirely to root macropores (Newman et al., 2004).

To investigate this hypothesis will require measurements in catchments to measure water flowpaths and residence times using tracers as well as fracture measurements, geophysical surveys and hillslope flow models. Time-intensive trench studies could also be completed (van Meerveld et al., 2015). Mapping roots and macropores will also be needed (Wu et al., 2014). In addition, a recent hypothesis suggests that the shallow lateral flowpaths underlying hillslopes in catchments are co-located at depth intervals marking biogeochemical reactions: in other words, the zones of lateral flow may be caused by or may mark the depth intervals where biogeochemical reactions have occurred over long time periods in catchments (Brantley et al., 2017). If that is true then a possible path forward would be to use drill cores or cuttings to identify geochemical reaction fronts in the subsurface and then use those to infer both pathways of vertical and lateral flow based on the geochemical signatures. Such an approach still must be tested with hydrologic models and measurements.

**4 Synthesizing Across Hypotheses and a Big Challenge**

As indicated previously, none of these building (H1-H4) and plumbing (H5-H9) hypotheses as summarized in Figure 1 are strictly architectural or strictly water-related, respectively. This intertwining is related to the actions of trees and water which are both physical and chemical in nature. For example, the exudates secreted by roots or their associated microbiota often chemically react with minerals (see hypothesis 2). Therefore, if roots penetrate

rock material (hypothesis 1), they make rock moisture more reactive. This in turn weakens the rock material and makes it more likely for the material to disaggregate (Bonneville et al., 2009; 2011). This is partly because propagation of a crack tip during disaggregation is essentially a breaking of chemical bonds and the ease of such a reaction increases when the tip is filled with more reactive fluid. Thus, tree roots and associated microbiota affect both the architecture and water chemistry.

As just described, the coupled aspects of tree-soil interactions related to architecture and plumbing are so tightly coupled that they can provide both positive and negative feedbacks. Another positive feedback is created by rhizospheres that develop around roots, creating macropores that channelize flow. This flow in turn produces higher densities of soil organic carbon and more intense nitrogen cycling which in turn promotes greater flow, more carbon, and more nitrogen cycling (Johnson and Lehmann, 2006). On the other hand, if all such feedbacks were positive in nature, development of regolith might be a runaway process. Implicit to hypothesis 3 is the idea that negative feedbacks must also be important so that thickness of mobile soil evolves toward a steady state.

If such a steady state can develop for mobile soils or even for the entire regolith, then some "telecommunication" is needed back and forth among processes at the top and processes at depth so that rates can balance. Most of the ideas as described by hypotheses 2, 3, 4, 5, 6, 7, and 8 (Figure 1) suggest that the CZ is shaped from the top down. For example, the ultimate top-down forcing factor may be dust as described in hypothesis 4. However, if fracturing ultimately controls the distribution of roots in unweathered rock (hypothesis 1), the CZ may alternately be shaped from the bottom up. For example, fracturing under hills has been posited to be controlled by the state of tectonic stress and how it interacts with topographic unloading (St. Clair et al., 2015). Such ideas suggest that distribution of trees and their access to water and the nature of the CZ may be ultimately dictated by bottom-up, tectonic controls.

Another proposed example of a bottom-up control on the CZ is drainage (Rempe and Dietrich, 2014). Rempe and Dietrich argue that the unweathered rock within a hill acts as the valve that controls drainage of water and the advance rate of weathering. Much work is needed to understand all the valves for water within hills (shown for simplicity as one valve in Figure 1). These valves partition water into evaporation, throughflow, stemflow, shallow lateral flow along perched saturated zones, matrix flow through the unsaturated zone to the water table, and ultimately flow to the channel. Hypotheses 5, 6, 7, 8, and 9 emphasize aspects related to how trees plumb some of these valves.

Perhaps one of the biggest hindrances toward forward movement in testing these hypotheses is that the different scientific communities do not speak the same language. Each discipline has terminology that does not transfer well from one discipline to another because of subtle connotations or denotations. For example, the depth of mobile soil to a geomorphologist is often very close in meaning to the depth of the primary rooting zone of the tree physiologist or the depth to the B horizon of the soil scientist or the depth to a reaction front as described by the geochemist. Likewise, macropores, rhizospheres, roots, and preferential flowpaths are not the same, but they all can sometimes refer to similar parts of the same system. Perhaps it is useful to point out that one aspect of this "naming" problem is that scientists that study the CZ try to define specific entities (such as layers) using operational definitions. In actuality, the CZ is the gradient defined by the changes in material equilibrated at depth as compared to material equilibrating to surficial conditions. All entities within the CZ such as layers shown in Figure 2 must be operationally defined because they are to some extent arbitrary depth intervals within a gradient of material properties. This is true for depth intervals as in Figure 2 as well as for types of water as shown in Figure 4: nomenclature is used to divide up somewhat arbitrary categories within the gradient which we call the CZ.

Perhaps the best (or only) way to break down the barriers created by terminology is to develop numerical models that integrate different concepts. This is difficult. As of yet, for example, tree root models are not incorporated into geochemical reactive transport codes for use in investigating the effects of roots on mineral-water weathering reactions. If such a model were available, water flow through macropores could be coupled with reactions stimulated within the rhizosphere. New models are also needed that incorporate concepts of connectivity and percolation or that move beyond continuum approaches to quantify weathering reactions at pedon, hillslope, and landscape scale.

## 5 Conclusions and a Vision for Moving Forward

The role of trees in building and plumbing the critical zone is poorly understood because the topic must be addressed by scientists of multiple disciplines trained to think in very disparate ways across vastly different timescales. Yet, understanding how soils form and are sustained is an important focus as the human population grows toward 10 billion in the next century. Soils act as natural filters of water but our understanding of the flowpaths and residence times of pore waters in forested soils are rudimentary. This paper has explored the role of trees as builders and

plumbers of the critical zone and the role of trees in the context of movement of water. Trees are the most important architects and plumbers of the CZ in many landscapes.

Much work needs to be done to understand the distribution of water content in the soil and the characteristic timescales of water movement and how it relates to trees. Similarly, research is needed to address how trees affect chemical, physical, and biological subsurface processes. Trees affect subsurface mixing and the movement of water in ecosystems (Figure 1), especially where the water that passes through a soil into a stream may be isotopically very different than the water that is held in that soil and taken up into the tree during transpiration (Figure 4). Such ecohydrological separation has implications for how we conceptualize and parameterize water storage and release in our models but a thorough understanding of these ideas requires understanding both the architecture of trees and the architecture of the critical zone (Figure 5). Groups of scientists must design and run initiatives to "map the roots", "map the fungi", "trace the water", and "model the tree and its soil" in the context of geochemical and soil variations, and the work must be focussed on settings where all disciplines can bring their tools of choice.

Observatory networks (Anderson et al., 2008; Banwart et al., 2012; Weathers et al., 2016; Brantley et al., 2017, in review) probably provide the only way to investigate all the chemical, physical and biological processes that are affected by trees. For example, the hypotheses stated here should be tested across the growing network of critical zone observatories. Alternatively, a few observatories could be chosen as a focus for tree observation. Likewise, global databases such as those for fine roots (http://roots.ornl.gov/), soil moisture (https://ismn.geo.tuwien.ac.at/), and sap flow (http://sapfluxnet.creaf.cat/app), could be used to extend or test hypotheses. Only with scientists crossing disciplines and studying the same sites together will questions be answered about how trees have plumbed and built the CZ. A focus on long timescales and the architecture of the CZ as investigated by geologists will elucidate the nature of short timescale water movements as studied by hydrologists and ecologists. Likewise, the interpretation of short timescale water movements will elucidate the nature of slow geological change at earth's surface. As humans impact the CZ more extensively and at more rapid rates, we will continue to need fundamental knowledge of both the long and short timescale phenomena that couple trees and the CZ.

**Acknowledgements**

This paper resulted from a workshop on Trees in the CZ funded by NSF EAR 13-31726 (PI: SL Brantley) and NSF ICER-1445246 SAVI: Crossing the Boundaries of Critical Zone Science with a Virtual Institute. The workshop

was facilitated by J. Williams, the Susquehanna Shale Hills Critical Zone Observatory, and Pennsylvania State University's Earth and Environmental Systems Institute. Authors were drawn from the 29 members of the workshop, representing 15 institutions and 8 Critical Zone Observatories. Other workshop members are acknowledged: H. Barnard, M. Green, C. Riebe, W. Silver, K. Brubaker, K. Davis, K. Gaines, Y. Zhang, L. Hill, Y. He, X. Gu, W. Zhi, and H. Kim. C. Bao is acknowledged for Figure 5 and L. Radville for help with Figure 1. H. Lin was consulted about macropores. D.L.K. acknowledges NSF EAR 1144760, S.A.P. acknowledges NSF EAR-1255013 and NSF EAR 1331408, J.A.M. acknowledges NSF-1452694, S.E.G. acknowledges NSF EAR 1331872, and D.M.E. acknowledges DOE-TES DE-SC0012003.

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

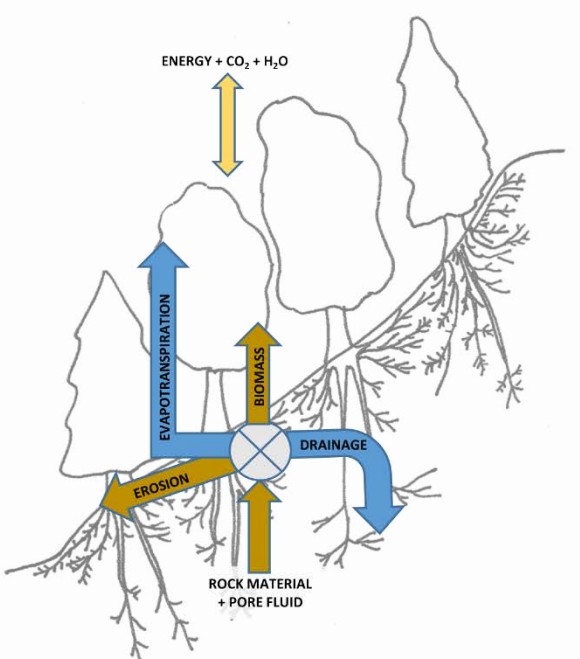

**H1 -** Tree roots can only physically penetrate and biogeochemically comminute the immobile substrate underlying mobile soil when that underlying substrate is fractured or pre-weathered.

**H2 -** In settings where the thickness of weathered material, $H$, is large, trees primarily shape the CZ through biogeochemical reactions within the rooting zone.

**H3 -** In forested uplands, the thickness of mobile soil, $h$, can evolve toward a steady state because of feedbacks related to root disruption and tree throw.

**H4 -** In settings where $h \ll H$ and the rate of uplift and erosion are low, the uptake of phosphorus into trees is buffered by the fine-grained fraction of the soil, and the ultimate source of this phosphorus is dust.

**H9 -** Even trees growing well above the valley floor of a catchment can directly affect stream chemistry where changes in permeability near the rooting zone promote intermittent zones of water saturation and downslope flow of water to the stream.

**H8 -** Mycorrhizal fungi can use matrix water directly but trees can only use this water by accessing it indirectly through the fungi.

**H7 -** Trees rely on matrix water in the unsaturated zone that at times may have an isotopic composition distinct from the gravity-drained water that transits from the hillslope to groundwater and streamflow.

**H6 -** Trees grow the majority of their roots in the zone where the most growth-limiting resource is abundant, but they also grow roots at other depths to forage for other resources and to hydraulically redistribute those resources to depths where they can be taken up more efficiently.

**H5 -** In settings of limited water availability, trees maintain the highest density of functional roots at depths where water can be extracted over most of the growing season with the least amount of energy expenditure.

**Figure 1: Trees transform energy + $CO_2$ + $H_2O$ (+nutrients) into biomass at the same time that they affect water fluxes, climate, erosion, weathering, hillslopes, distribution of elements and microbiota in soils. Nine hypotheses are proposed about these inter-relationships for future testing. As energy from the sun radiates on to the earth at about 800 watts $m^{-2}$, trees act like power plants that transform energy (into biomass) and flush water (transpiration). A single tree can transpire on the order of 100 kg water day$^{-1}$. The trees and their roots are shown with the symbol for a valve ($\otimes$) to emphasize that trees act to partition water into the atmosphere (evapotranspiration), into throughfall, into stemflow, and into the subsurface where water can flow along roots and macropores (see text). At the same time that water is removed from soil and transpired, tree roots embed themselves in the soil and stabilize its structure. As the tree and its associated microbiota inject acids and other exudates into the soil, nutrient material is solubilized, taken up into the tree, and then returned to the soil after the leaves fall or the tree dies. Likewise, after dying on hillslopes, tree fall can lift the rock material in the root wad, moving it toward the earth's surface and then downhill. Over much longer timeframes, such bioturbation moves soil downslope. In these ways, trees act as stirring agents, moving nutrients and particles from rooting depth to land surface through chemical and mechanical processes, respectively.**

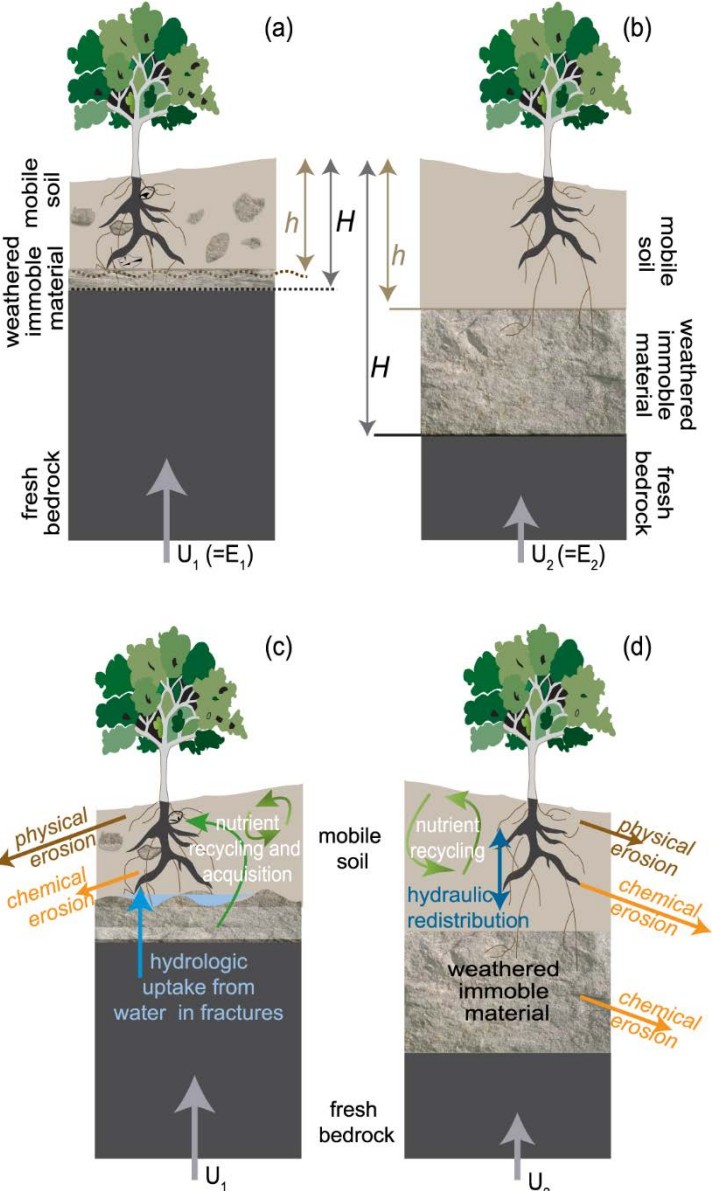

**Figure 2. Conceptual model for two end members of steady state forested profiles such that uplift rates (U) equal erosion rates (E): 1) left side (2a and 2c) where trees profoundly influence architecture and plumbing processes in the critical zone ($h \cong H$) and 2) right side (2b and 2d) where trees may amplify or modulate critical zone processes; however, they do not influence the deeper architecture ($h \ll H$). Upper figures emphasize architectural differences whereas lower figures describe differences in processes and erosion rates. We posit that the ratio of the thickness of the mobile layer ($h$) to that of the entire layer of weathered material ($H$) -- both immobile and mobile -- is set by the balance between erosion and weathering processes. When $U$ and $E$ are rapid (i.e., $U_1 > U_2$; $E_1 > E_2$), tree roots not only set the boundary between the mobile and immobile layers, but through growth and turnover can impart a 'wavy' boundary, and can inject detached fresh rock and mineral material in a range of sizes into the mobile soil layer by wind sway, growth-driven root actions, and tree throw (2a). This contrasts with a slower uplift and erosion rate (e.g. $U_2$) where roots are predominantly contained within the mobile soil layer, the interface between immobile and mobile material is generally less wavy, and grains of material injected from below into the mobile soil are generally finer and more weathered than in a fast-eroding setting (Fig. 2b). When the $h/H$ ratio $\cong 1$, physical erosion likely dominates over chemical erosion, both of which are restricted essentially**

to *h*. In this regime, root fungi acquire nutrients from both recently detached grains in the mobile layer and, to a lesser extent, from fresh bedrock (2c). In contrast when *h* <<*H*, chemical erosion dominates in both the mobile and immobile layers and root fungi are restricted mainly to merely recycling material within the mobile soil layer, with only a small influx of nutrients from the much lower density of roots extending into the deeper immobile material below (2d). The difference in architecture potentially influences

subsurface hydrologic routing and storage: when *h* ≅ *H*, the wavy interface at the boundary of mobile and immobile material promotes opportunities for 'fill and spill" (water ponded in depressions as shown in blue), while fractures store water that is accessible for root uptake. In contrast, when *h* exceeds the depth of penetration of most tree roots as in (2d), the architecture may not promote opportunities for "fill and spill' nor for water in fresh bedrock to be an important as a source for trees. While hydraulic redistribution could happen in both end members, we show it in (2d) to emphasize that most roots in this end member do not access

fracture-held water in fresh bedrock.

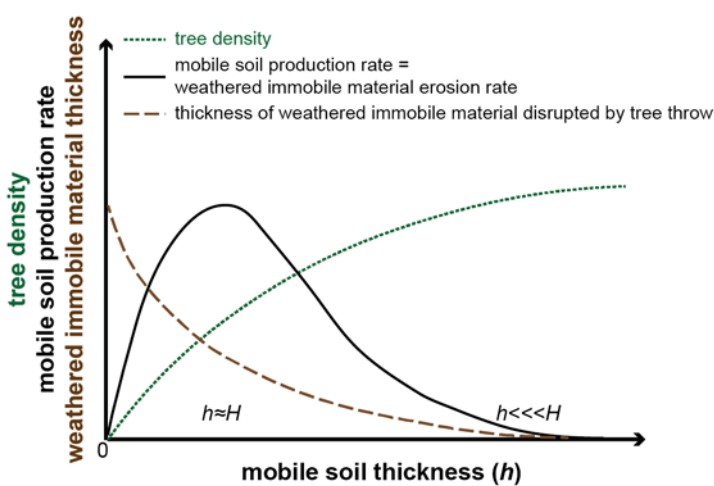

Figure 3. Conceptual relationship proposed by Gabet and Mudd (2010) showing i) tree density (green dotted line), ii) production rate of mobile soil (black line), and iii) thickness of weathered immobile material that is disrupted by tree throw (brown dashed

line), all plotted as a function of the mobile soil thickness. As shown, the tree density and the thickness of weathered immobile material disrupted by tree throw events are thought to vary with thickness of the mobile soil (*h*). With increasing soil thickness, the tree density increases while the thickness of immobile material disturbed during tree turnover decreases. Over geomorphic timescales, the mobile soil production rate is inferred here to equal the product of tree density times thickness of bedrock disrupted by each tree throw times tree throw frequency (not shown). In a steady state landscape, this mobile soil production rate is equivalent

to the weathered immobile material erosion rate. The rate first increases and then decreases because thin soils support too few trees to create mobile soil from immobile material at a significant rate but thick soils insulate underlying immobile material from significant root disturbance. We hypothesize that maximum soil production by tree throw occurs when the thickness of mobile soil (*h*) ≈ thickness of all weathered material (*H*).

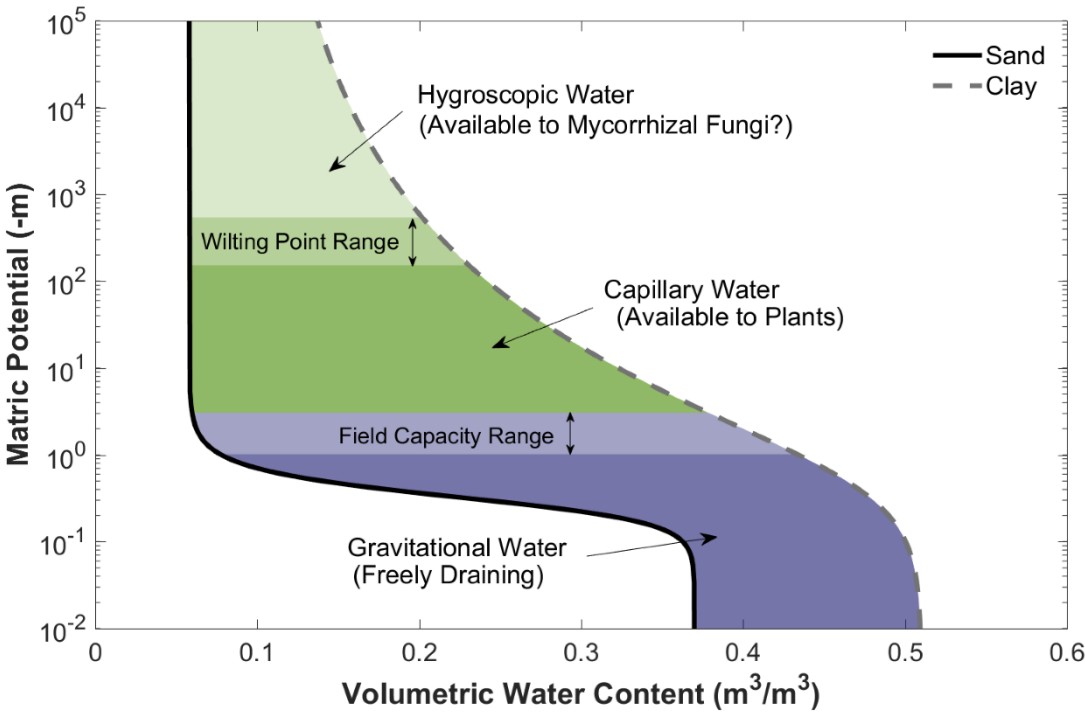

**Figure 4. Water available to streams, trees, and mycorrhizae may come from pores that drain under different tensions. Only water that is freely draining will contribute to streamflow whereas matrix waters, held at tension in soils or rock, will not. Matrix waters include capillary waters available to plants, and hygroscopic waters that are held at tensions beyond the wilting point (and thus unavailable to) agronomic plants. Such waters may be available to mycorrhizal fungi (see hypothesis 8). More energy is required to acquire water that is held under higher tensions, so we hypothesize that plants will use water that is most energetically favorable (hypothesis 5).**

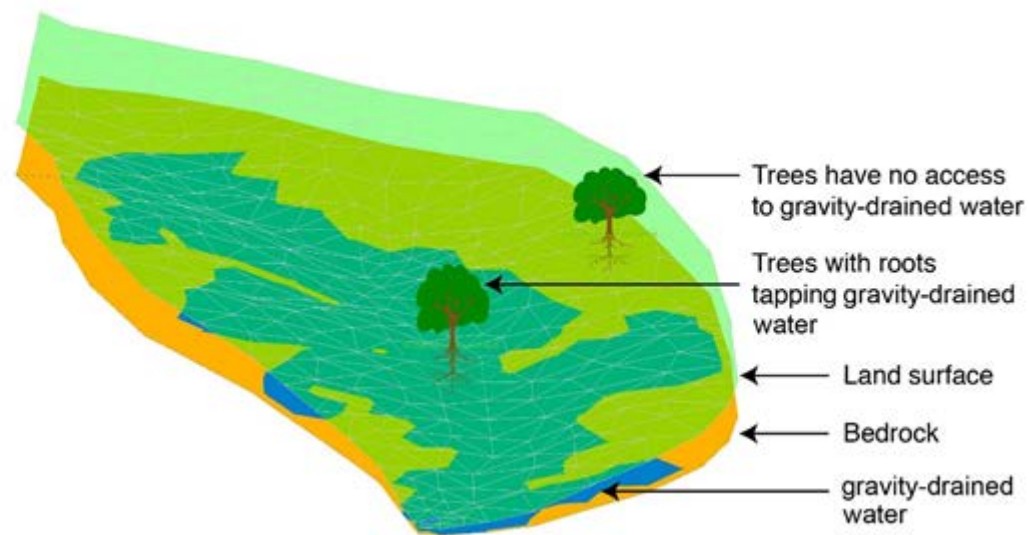

**Figure 5. A schematic diagram showing how connectivity of a landscape might affect the distribution of water that is drained by gravity or held in the matrix. Gravity-drained water enters as rainfall, drains vertically through the unsaturated zone to the groundwater, and leaves the watershed. Under this scenario, roots of trees high in the catchment do not access this water (except ephemerally during drainage), and instead may rely predominantly on matrix water. These trees may not have roots that reach the groundwater because of the thickness of the unsaturated zone and weathered material high in the landscape. By contrast, roots of trees in the channel or swales may access gravity-drained and matrix water as well as the bedrock interface and groundwater. This diagram emphasizes that trees high in the watershed may not interact with the stream because of low connectivity. In contrast to this conceptual picture, many watersheds may have intermittent connectivity between trees high in the catchment and the stream because of transient saturation at the bottom of the rooting zone or at the interface of mobile soil and the underlying weathered immobile material. Such transient perched water tables may allow down-hillslope flow of water from the ridgetops to the stream, providing intermittent connectivity (see hypothesis 9).**

**Table 1. Nomenclature**

| Name of layer | Description of earth material in layer | Description of trees in the layer |
|---|---|---|
| Fresh bedrock | Parent material that has not been affected by surface processes (R layer in soil sciences or protolith in geology). Fresh bedrock is unweathered and typically underlies weathered immobile material. | No tree material present |
| Weathered immobile material (thickness = H-h) | Material commonly denoted as C layer in the soil sciences which has been chemically altered but manifests the fabric of the fresh bedrock. The extent and distribution of weathering is influenced by fractures and other structural properties of bedrock. This zone can contain weathered rock, saprolite, and/or saprock. Overlies fresh bedrock. | This zone contains tree roots, which may enhance physical and chemical weathering through root expansion, mineral acquisition including that of mycorrhizal fungi and uptake or release of water. |
| Mobile soil or colluvium (layer defined to have thickness h) | Mixed, disrupted or churned material which contains mineral and organic constituents. Mobile soil reflects displacement from the original bedrock fabric (but not necessarily significant transport) via detachment, mixing, or larger-scale transport (e.g. via ice lens growth, gopher burrowing or tree throw) such that the fabric of the original bedrock is no longer intact, and the material is available for transport. This contrasts with H, which is the depth that encompasses both immobile and mobile weathered material. | This zone, which contains most of the tree roots, is the zone most chemically influenced by trees. Woody roots (including tap roots when present) typically can reach below this zone. |
| Type of water | Description of water | Other terms used |
| Gravity-drained water | Water that flows freely under the force of gravity. | Also referred to as "mobile" water or "freely drained" water. |
| Matrix water | Water that does not flow freely under gravity and is composed of hygroscopic and capillary water. Capillary water consists of water held at tensions greater than the agronomically-defined wilting point, and water between the "wilting point" and field capacity. Hygroscopic water forms thin films around soil particles, held at tensions beyond the wilting point of agronomic plants. | Also commonly referred to as "immobile", "bound" or "tightly bound" water. "Matrix water" is preferred here because tightly bound water may not be immobile over timescales relevant to CZ researchers. |
| Types of fungi | Description of fungi | Other terms used |
| Arbuscular mycorrhizal fungi (Van der Heijden et al., 2015) | Fungi belonging to the Glomeromycota that colonize most herbs, grasses, tropical and many temperate trees. These fungi colonize inside the plant cell of absorptive roots and are most noted for their ability to improve acquisition of phosphorus and other relatively immobile nutrients. AMF include an estimated 300-1600 fungal taxa colonizing about 200,000 plant species. | AMF |
| Ectomycorrhizal fungi (Van der Heijden et al., 2015) | Fungi belonging to Basidomycota and Ascomycota that colonize trees in the pine family, Eucalyptus, oaks, beech, birches and many other temperate and boreal trees. These fungi colonize root tips and do not enter the plant cell. They are able to more readily use organic forms of nitrogen and phosphorus than AMF and their hyphae can fuse to form long, relative thick strands called rhizomorphs, eventually leading to mycelial mats in the forest floor. EMF include an estimated 25,000 fungal taxa colonizing ~6000 woody plant species | EMF |