# Peer review of "Reviews and syntheses: On the roles trees play in building and plumbing the critical zone"

_Biogeosciences, 2017_

## Referee Comment (RC1) · L. L. Taylor (Referee) · 10 Apr 2017

General comments:

This manuscript presents a set of cross-disciplinary hypotheses addressing the impact of trees on the Critical Zone over a wide range of time and spatial scales. As the different disciplines (eg geology, ecology, soil science) have taken very different approaches to these questions, there is a need for all of us both to revisit "established" viewpoints and to consider new ones in order to gain a holistic understanding of the function and development of the Critical Zone. Amongst the hypotheses are several which I have been thinking about for some years, and I am pleased to see them presented here along with several other things I had not thought about in as much detail before. This paper provides a useful framework for cross-disciplinary cooperation.

[Figure]

Specific comments (explicit list of things to do at the end above "Technical corrections"):

Hypothesis 1. I have long been rather skeptical of claims that roots (or hyphae) exert forces that fracture bedrock, especially in directions perpendicular to the bedrock surface where there is nowhere for the rock to go (eg a free surface as on a hillslope, or into compressible soil). What is the role of the chemical environment of a root or hypha (ie the rhizosphere or hyphosphere) in crack propagation?

Hypothesis 2. I think that dissolution rates for surfaces covered with cryptogams (lichens and bryophytes) could do with more investigation, as the critical zone for these ecosystems is very different from those with vascular plants.

Hypothesis 3. Is tree throw a function of tree density? Do trees fall less frequently when surrounded by other mature trees? What is the slope angle of of the Oregon hillslopes discussed in lines 16-29, for comparison with that of West et al 2013? What was the state of the canopy in the forest on shale studied by West et al? Does this forest have pit-mound features? What is the disturbance timescale for these sites, and do either of them have smectites that might lead to more frequent landslides? Where do steady-state values of "h" occur?

Hypothesis 4. This section needs a brief discussion of how dust particles differ from soil particles. Does dust have a smaller anion exchange capacity? Is dust primarily mineral (and if so what sort of minerals) or is there an organic component? Why would dust be a better source of P than soil? Are the particles so small that they disappear and leave a legacy of ions behind?

Hypotheses 5,7,8. It sounds like we need a thorough overview of wilting points for a variety of plant and mycorrhizal functional types; this has important implications for global vegetation models especially if they use hardcoded or database values for wilting points. Is the greater amount of nutrients in surface soils due to organic matter; is this because the plants are in general recycling nutrients? Are the deeper, saturated soils referred to on line 15 page 16 above field capacity, and are the roots aerenchymatous?

It may be worth citing Bornyasz et al 2005b under hypothesis 8, as they showed that ectomycorrhizal hyphae penetrated bedrock matrix. They cited Hubbert et al 2001b and Egerton-Warburton et al 2003 as earlier sources suggesting that hyphae were important for tree water relations; these should probably be cited along with Bornyasz et al 2005b.

Hypothesis 6. Hydraulic redistribution of nutrients must depend on the amount of nutrients that enter the roots in the transpiration stream and are then not taken up internally. How well understood are those processes? Also, is there an effective redistribution of nutrients due to throughfall, ie leaching of ions from leaves by rainwater, and if so, it this a function of leaf attributes such as shape, specific leaf area, etc?

Hypothesis 9. The effect of different soil layers on streamwater chemistry is surely strongly dependent on the timescale of interest. It begs the question of how well we understand the residence time of water in the different soil compartments. Is there a reference available for the statement that most streamwater solutes originate from soil weathering?

General. As stated in the conclusions, the characteristic timescales of water movement are critical for understanding the Critical Zone. I am a bit surprised that there was no explicit discussion of macroporosity however, as old roots are important components of macroporosity and conduits for water to deeper layers. Also, not all sections provide any suggestions for how to test the hypotheses, or what needs to be done, ie do we need to collect a lot of data, do we need new techniques, is there conflicting evidence etc. However, given the wide range of material covered, I think this is a good paper.

Things that the authors should do: Most of my comments above were questions that came to mind because I was interested rather than because I think the authors need to make major changes (although they can certainly do so if they wish). There are however a few things that need to be done: 1. Address the questions about the field sites discussed in Hypothesis 3, ie so that comparable information eg slope angle is

given for both. 2. Answer the questions about why dust is a better source of P that soil particles under hypothesis 4. 3. Add some citations for hypothesis 8 as described. 4. If there is a citation for the the statement that most streamwater solutes originate from weathering in soil, please add it.

Technical corrections. This paper is generally well-written, but I do have a few corrections to list.

Terminology. The manuscript divides the regolith into three units: mobile soil, weathered but immobile material, and fresh bedrock. This is fine, but there are a few places in the manuscript where it is not immediately clear which of these three units is being referred to. For example, under Hypothesis 3, page 12, line 3, and also page 13, line 21, one needs to refer to Figure 3 to know that "soil thickness" refers to "h" (mobile soil depth). As Figure 3 is discussed here that is perhaps OK, but on page 20, Hypothesis 9, line 7, it is less clear. Weathering reactions occur in both the mobile and immobile soil but which of these layers is the primary donor of weathering products to streamwaters? Please clarify.

Other corrections and suggestions: 1. When at first reading the abstract (line 24), it was not immediately clear to me whether "the depth of weathered material, H" referred to the top or the bottom of the weathered layer. On line 25 it is perhaps implicit that "h" is measured from the soil surface. However, one's familiarity with these terms depends on one's disciplinary background. 2. Define "denudation" which occurs in section 1.3, page 8, line 12. I have seen this term used in different ways, to express physical, chemical or all removal of material. 3. Define "soil dilation" which occurs in section 2.3 page 11 line 28. 4. Page 14 Section 2.4 line 27: The sentence "These long time periods can magnify slow losses of P." sounds slightly odd; I had to read it a few times. It means that P losses slow down as rates of uplift and erosion are slow, but perhaps it could be rephrased... 5. Figure 1. There are four panels including three photographs that are not described. These photos need some sort of reference or description: where are they, what are they meant to illustrate here? Also, the labels on the conceptual

diagram have lost their final letters: EVAPOTRANSPIRATIO (with N having wrapped around) and DRAINAG (with E having wrapped around). 6. Figure 2. Panels C and D are never referred to. They clearly belong with Panels A and B respectively and are implicitly discussed as such, but they need references of their own if they are to be labeled separately. also, they contain some awkwardly-phrased labels. "nutrient aquire" should be "nutrients aquired" or "nutrient aquisition", while "nutrient recycle" should be "nutrients recycled" or "nutrient recycling". The panels could be larger; they could easily fill the width subtended by the caption. 7. Figure 3. I had to look at this figure for a while, as I was thrown by the labels h∼H and h«H. H is not defined anywhere in the figure or its caption. Perhaps label the X axis "mobile soil thickness" and somehow relate H to the thickness of bedrock disrupted by tree throw. I am still not sure how these are related, and maybe it is just not clear anyway; I realise this is someone else's figure. Once I had read the text I understood what was meant (mostly), but I looked at the figures before reading the text and so will many of your readers. 8. Table 1. "is comprised of" is not correct... replace with "is composed of" or rephrase.

---

## Referee Comment (RC2) · Anonymous Referee #2 · 14 Apr 2017

Review of bg-2017-61

Overview: I read with great interest this review manuscript on the role of tree roots in the Critical Zone (CZ). I applaud the authors for taking on this enormous task of bringing together a diverse set of disciplines involved. The result is a tour de force. The manuscript sets up the role of tree roots and associated mycorrhizal fungi within two broad categories of 1) building and 2) plumbing the CZ. My overall assessment is this is a good start on development of a review and synthesis manuscript here. But in its current form the emphasis is big R (review) and little s (synthesis), and the manuscript would be stronger if there was also big S. There is quite a bit of variance in the depth of insights provided among the hypotheses. None of them give clear paths forward for developing the science. I think the authors should make bold statements here. After all, the aim of review/synthesis manuscripts is to identify new knowledge where it is not

obvious from the individual parts.

Specific comments: I read this three times, and each time I tried to think about the distinction between the two broad categories. It remains unclear how building the CZ is distinct from plumbing it. The hypotheses do not help clarify these categories either, and in some cases they add confusion. Some of them seem immediately testable, at least with the effort to get the samples in space or time (e.g., H1, H5), while others are general, somewhat vague propositions (e.g., H2, H6). One suggestion is to turn these into questions, which would lend a more uniform treatment to the sub-categories. A second issue with some of the hypothesis sections is that they read a little like watching a tennis match. For example, in one section roots use shallow water, except when they do not. In another, roots use water that is isotopically similar to rainwater, except when they do not. These are clearly opportunities for identifying research questions that could make a huge difference, but they are not used that way within the hypothesis sections, and so there are many missed opportunities. A third issue with the hypotheses is that there is little synthesis across them. There are a few hooks here and there, but less effort to weave them together so that readers can see the new knowledge to be gained. Fourth, the hypothesis sections never highlight the challenges or issues to be resolved, or tell the reader how to advance science. What are some next steps? Some attempt at synthesis and advancement is started in the conclusions section, but I think this has to be done within each hypothesis section and then tied together in the conclusions.

Specific comments: Title: The title is too broad. The manuscript is focused on the role of tree roots and mycorrhizal fungi on CZ processes. With the exception of a brief mention of the differences in root traits among AM and EM species the manuscript does not make much of tree traits in general, and so I think the title is currently a bit misleading.

Figures: Some of the figures are either no necessary or are a bit misleading. Figure 2 is important, but some of the details are not provided in the caption. In particular, terms

such as h and H are not defined here, and graphics C and D are not described. Figure could be deleted without harming the paper. It is barely mentioned in the text and its portrayal of trees with roots accessing gravity-drained water or not oversimplifies the problem.

References: There are a number of errors in the references, including duplicates (e.g., Bornyasz et al 2005a,b), incorrect issue number (e.g., Chen et al., 2013 PNAS), and wrong journal (e.g., Jackson et al., 1999 should be PNAS, not Ecology).

---

## Short Comment (SC1) · 27 Apr 2017

General Comments:

This manuscript provides an engaging review of the science regarding the role of tree roots, and associated mycorrhizal fungi, in moderating processes within the critical zone. The topic is timely and relevant to a broad readership, especially (as noted by a previous reviewer) if the synthesis components can be further emphasized to meet the high level of the review components and together work to help steer directions forward within the research community. The paper is well written, and previous reviewers have provided detailed comment on the text and figures that address many of my concerns. However, I believe the figures are in need of additional improvements beyond what has already been highlighted by the other reviewers. Several unaddressed concerns still

remain and have inspired myself, and my affiliated research group, to provide comment to help in improving this important piece. These comments are both my own, and, in-part, are on behalf of the Southern Sierra CZO, and were gleaned through a lively discussion about this paper orchestrated during a recent research group meeting.

Specific Comments:

1) Figure 2: In the Southern Sierra CZO and Reynolds Creek CZO, as well as other locations, it is observed that roots are commonly emplaced within the weathered im-mobile material, growing and accessing the water there (Arkley, 1981; Graham et al., 2010; Niemeyer et al., 2017). Tree roots however are not drawn within the "weathered immobile material" in Fig. 2B (they are only drawn down to its top surface). Addition-ally, is there a reason why the mobile soil is deeper in Fig. 2B than Fig. 2A? I agree "H" can be significantly deeper in some locations/regions more than others, but for a similar slope with similar vegetation sizes I would expect "h" to be similar, despite large differences in "H". Regarding figure details, previous reviewers point out some gram-matical issues, but additionally in Fig. 2D is the "hydrologic pump" arrow intended to point the other way?. Also, the "weathered immobile material" in Fig. 2D is missing its cross-hatching in comparison to Fig. 2B.

Arkley. (1981). Soil Moisture Use by Mixed Conifer Forest in a Summer-Dry Climate. SOIL SCIENCE SOCIETY OF AMERICA JOURNAL.

Graham, R. C., Rossi, A. M., & Hubbert, K. R. (2010). Rock to regolith conversion: Producing hospitable substrates for terrestrial ecosystems. GSA Today, 20(2), 4–9. Journal Article.

Niemeyer, R. J., Heinse, R., Link, T. E., Seyfried, M. S., Klos, P. Z., Williams, C. J., & Nielson, T. (2017). Spatiotemporal soil and saprolite moisture dynam-ics across a semi-arid woody plant gradient. Journal of Hydrology, 544, 21–35. http://doi.org/10.1016/j.jhydrol.2016.11.005

2) Figure 3: In the legend box, "Bedrock erosion rate" could be more accurately described as "Weathered immobile material erosion rate" or perhaps "Saprolite erosion rate" (if the authors are comfortable with that nomenclature). This detail is buried within the caption, but I feel this needs to be clearer in the figure itself, especially when in Fig. 2 the word "bedrock" is only used in context of fresh bedrock. I realize this figure is based on a previous figure, but these types of changes to the figure's text should be easy to make and can still pay reference to the original figure. Similarly, in the legend "thickness of bedrock" could be altered to "thickness of immobile weathered material" or "thickness of saprolite", and perhaps "depth of saprolite disrupted" for the Y axis. Using "Bedrock thickness" invokes the thought of fresh bedrock and is confusing to the reader. I agree with the hypothesis conceptually if these changes are made, but it needs to be clear that tree throw is only going to be removing material around its large roots, which likely do not extend all the way to fresh bedrock (as it looks like it is drawn in Fig. 2A), but instead are eroding weathered material at the mobile/immobile boundary only.

3) Table 1: For the readers unfamiliar with mycorrhizal fungi nomenclature, it may be useful to add a section on "Types of fungi", or something similar, to Table 1 to help in the readers in understanding potentially new terminology relevant to the manuscript (e.g AM/EM differences, etc.).

Technical Corrections:

1) Figure 1: I agree with a previous reviewer that this figure may need more explanation (and typos fixed) if it is kept in. If it is kept, I believe the pictures of roots need to have the roots highlighted in some way. As it is now, it is hard to discern the roots within the dark photographs. In general I don't believe this figure adds much to the discussion and perhaps could be left out without taking away from the main ideas of the piece.

---

## Author Comment (AC1) · 24 May 2017

Response to "Reviews and syntheses: On the roles trees play in building and plumbing the Critical Zone" We appreciate the insightful questions and comments we received on our paper, "Reviews and syntheses: On the roles trees play in building and plumbing the Critical Zone" from L. L. Taylor, P. Zion Klos, and an anonymous reviewer. We would like to revise our paper to take into account the points that were made. In doing this we will seek to even out some of the treatments among the hypotheses. The reviewers made many small points and posed many small but pertinent questions that we can address throughout (about slope angle, dust properties, citations, etc.). On the other hand, many of the questions mentioned by the reviewers already show how our hypotheses are stimulating questions for future work (questions about biogeochemical impacts, hydraulic redistribution, and others). Below, we discuss the reviewers' more

[Figure]

general comments that we have binned into categories.

First, we welcome the insights from several of the reviewers about the title of our paper. Perhaps this would be a better title: "Reviews and syntheses: On the roles tree roots and mycorrhizal fungi play in building and plumbing the Critical Zone". We still argue that we need both "building and plumbing" because we want to emphasize that the growth of weathered material and soil from bedrock and the functioning of this part of the Critical Zone is very much affected by trees in terms of both physical (building) and chemical (plumbing) processes. Of course, "building and plumbing" are metaphors for processes that are not mutually exclusive – nor do they emphasize the many biological parts of the processes – but we think the words give the reader the sense of the paper in a short and succinct title. Our paper is meant to focus attention on the need to develop conceptual and numerical models that yield better understanding of how trees impact the architecture of the Critical Zone.

Second, we received many comments about Figure 2 and about the nomenclature for mobile soil, weathered immobile material, and fresh bedrock. Although it is not our intent to argue too much about nomenclature, we now realize we need to go through the manuscript carefully and reduce ambiguity by using only the three defined names for layers throughout. In choosing these three names we were trying to solve the problem (at least in this paper) of different definitions among different disciplines, among different countries, and even among different parts of individual countries. Obviously this part of the Critical Zone represents a gradient from ambient conditions at the land surface to deeper-earth conditions at depth, and gradients cannot always be sub-divided easily into layers. We also see that we need to make some changes in Figures 2 and 3 as suggested by the reviewers with respect to these naming conventions and concepts.

Third, the reviewers ask for more synthesis. We agree that we need to do a better job in the Conclusions wrapping together some of the implications that cross-cut the hypotheses. For example, we propose to emphasize the relationship of roots with preferential flow (tying together H1 and H9) and elucidate the inter-relationship of dissolution and

cracking (tying together H1 and H2). Furthermore, we agree with Taylor that lack of discussion of macropores is a major oversight. Such a discussion would fit in very well with H9 where we could discuss the idea of vertical and horizontal macropores and how these features inter-connect – and how they can be influenced by tree roots. Likewise, we feel that the importance of stress corrosion cracking – the phenomenon where corrosive fluids hasten the propagation of cracks in rocks – has been under-emphasized in the Critical Zone literature and could be amplified in the current paper, as requested. Addressing such topics in the conclusion will be used to synthesize the hypotheses.

Fourth, one reviewer asked for a re-phrasing of our hypotheses as questions. We resist that idea because we would lose clarity and because questions tend to multiply so quickly, while hypotheses are difficult to phrase (so they do not proliferate so easily) and are also instructive to test. On the other hand, testable hypotheses do demand experiments, as discussed below.

The one last overarching request by the reviewers is a roadmap for the future. We originally wrote a section with a summary of some proposed experiments. But such a set of experimental strategies is not that easy to design when communicating across disciplines and when problems remain undefined. Ultimately, the section on proposed experiments that we wrote felt like an add-on to us: we felt that the real meat of our work had been accomplished by defining the questions.

Thus, we decided that it might be beyond the scope of this paper to put together an experimental roadmap: the paper is already long, and the roadmap is not clear. We think, however, that the reviewer may have pointed a way forward for both synthesis and roadmapping: several of the hypotheses can be tested by experiment (H1, H2), others can be tested by identifying samples or mapping across space and time or by modelling (H3, H4, H5, H6), while others may require careful comparisons among techniques of water extraction (H6, H7) or use of tracers (H7, H8, H9). Perhaps a discussion along those lines would be an appropriate way to not only point to the future

but also tie the paper together. This is the response we propose. However, if the editor requests a section with experiments (and a longer paper), we are also happy to outline the specific CZ experiments we had originally discussed in an earlier, longer draft of this manuscript. We are happy to proceed on either approach after guidance from the editor.

―――――――――――――――――

---

## Author Response (AR1)

Response to "Reviews and syntheses: On the roles trees play in building and plumbing the Critical Zone"

We appreciate the insightful questions and comments we received on our paper, "Reviews and syntheses: On the roles trees play in building and plumbing the Critical Zone" from L. L. Taylor, P. Zion Klos, and an anonymous reviewer. We revised our paper to take into account the points that were made. In doing this we evened out some of the treatments among the hypotheses. The reviewers made many small points and posed many small but pertinent questions that we can address throughout (about slope angle, dust properties, citations, etc.). On the other hand, many of the questions mentioned by the reviewers already show how our hypotheses are stimulating questions for future work (questions about biogeochemical impacts, hydraulic redistribution, and others). Below, we discuss the reviewers' more general comments that we have binned into categories.

First, we welcomed the insights from several of the reviewers about the title of our paper. We decided not to change the title of the paper. We still argue that we need both "building and plumbing" because we want to emphasize that the growth of weathered material and soil from bedrock and the functioning of this part of the Critical Zone is very much affected by trees in terms of both physical (building) and chemical (plumbing) processes. We discuss how building and plumbing are intertwined (in abstract, in the first section) Of course, "building and plumbing" are metaphors for processes that are not mutually exclusive – nor do they emphasize the many biological parts of the processes – but we think the words give the reader the sense of the paper in a short and succinct title. Our paper is meant to focus attention on the need to develop conceptual and numerical models that yield better understanding of how trees impact the architecture of the critical zone. We elected to not add "roots and fungi" because we felt that this added complexity in the title did not really add that much after all. "Building and plumbing" certainly connotes roots and fungi, but we discuss many aspects of how trees evapotranspire water and change water residence times and flowpaths that are not strictly related to roots and fungi.

Second, we received many comments about Figure 2 and about the nomenclature for mobile soil, weathered immobile material, and fresh bedrock. Although it is not our intent to argue too much about nomenclature, we went through the manuscript carefully and reduced ambiguity by using only the three defined names for layers throughout. In choosing these three names we were trying to solve the problem (at least in this paper) of different definitions among different disciplines, among different countries, and even among different parts of individual countries. Obviously this part of the critical zone represents a gradient from ambient conditions at the land surface to deeper-earth conditions at depth, and gradients cannot always be sub-divided easily into layers. We made some changes in Figures 2 and 3 as suggested by the reviewers with respect to these naming conventions and concepts. We have also significantly clarified the caption for Figure 3, including making it clear what the difference is between 2a and 2c versus 2b and 2d.

Third, the reviewers ask for more synthesis. We added discussion in a new section (4 Synthesizing Across Hypotheses and Big Challenges) and we expanded the conclusions a bit. In addition, throughout the paper we emphasized the relationship of roots with preferential flow (tying together H1 and H9) and elucidate the inter-relationship of dissolution and cracking (tying together H1 and H2). Furthermore, we agree with Taylor that lack of discussion of macropores is a major oversight. We added discussion of macropores in several places but mostly in H9. We discussed the idea of vertical and horizontal macropores and how these features inter-connect -- and how they can be influenced by tree roots. We also mentioned the importance of stress corrosion cracking – the phenomenon where corrosive fluids hasten the propagation of cracks in rocks.

Fourth, one reviewer asked for a re-phrasing of our hypotheses as questions. We resisted that idea because we would lose clarity and because questions tend to multiply so quickly, while hypotheses are difficult to phrase (so they do not proliferate so easily) and are also instructive to test. On the other hand, testable hypotheses do demand experiments.

The one last overarching request by the reviewers is a roadmap for the future. In this new version of the paper, we tried to add in some ideas for approaches within discussions of each hypothesis. We also added in a brief section into the Synthesis section and the Conclusions section suggesting a few ideas for initiatives for the future. Such a set of experimental strategies is not that easy to design when communicating across disciplines and when problems remain undefined. We decided that it might be beyond the scope of this paper to put together an experimental roadmap: the paper is already long, and the roadmap is not clear. We do emphasize that communication is one of the big problems and we point out that different disciplines have different words for the same things (and we give examples). We need numerical models to clear up these confusions. We also emphasize the need for observatories where all disciplines work together. This is the path forward.

Point by point : L. L. Taylor

Hypo 1. We have now discussed stress corrosion cracking briefly in the manuscript.

Hypo 2. We mentioned exudates and their effect on stress corrosion cracking and weathering.

Hypo 3. We have amplified the caption of Figure 3 and text that addresses these questions. We added info in about slope and pit mounds for Oregon and PA. We explicitly mention that it is unknown whether steady state systems occur.

Hypo 4. We added a sentence explaining how dust differs from soil particles and why dust can be a better source than soil.

Hypo 5, 7, 8. We added a citation to Bornyasz.

Hypo 6. The questions here are beyond what we know!

Hypo 9. We agree that residence time of water is important and we mention that in several places in the manuscript. We don't know of a paper saying most stream solutes derive from soil weathering. (But where else would solutes come from other than atmospheric).

General. We added in a large discussion of macropores. That was an oversight on our part. We also tried to add in ideas for approaches for each hypothesis. This was a very good idea to discuss macropores more thoroughly.

Terminology. We went through every place the reviewer pointed out our terms were confusing and made a clarification.

Other corrections.

1. We have tried to make H and h very clear throughout.

2. We now define denudation.

3. We removed dilation.

4. We rewrote the offending sentence to make it more clear.

5. We removed photos. We fixed typos.

6. We extended the caption.

7. We revised this figure and made it more clear.

8. We removed "is comprised of"

Anon reviewer 2

We address most of these comments above in the general statements. We tried to provide more synthesis. I think we are still lacking but this is really hard! We feel like our paper is a launching pad for the synthesis that will happen in the next ten years. The science will be the synthesis. We need the science. We appreciated the kind words and thoughtful comments.

We tried to fix all the references.

P. Zion Klos

We tried to emphasize synthesis to the extent we could do so.

Specific comments

1. We now show some roots in the weathered immobile material in figure 2. We think in general, when h << H, the mobile soil is likely to be thinner and so that is why we made the figure that way. However, the difference is now subtle in the figure. We fixed the arrows in the figure as well. We have tried to make the different layers look similar in each panel.

2. We have revised the legend as requested to make it more clear. We think the figure now does everything that the reviewer requested.
3. We added in fungi to the table.

Technical Corrections

1) We have tried to make the explanation of this figure more clear and we removed the photos. We have amplified the discussion of trees as "valves" which is an important concept for the paper. This figure is important because it sets the stage for the paper.

[revised manuscript text omitted]

---

## Referee Report (RR1)

**On the roles trees play in building and plumbing the critical zone**
**Final minor corrections**

**Page 2 line 9: insert a space between e.g. and the reference Dokuchaev.**

**Page 2 Lines 17 and 18: In the sentence:**
*As such, a specific focus is on trees -- the most successful entities...*
**Include the word "terrestrial".** Marine productivity can be higher according to Chapin et al 2011 fig 6.5 and table 6.4. Trees are the most successful terrestrial entities... Also, **replace the two hyphens with an em-dash (throughout the manuscript; there are 10 instances and these are not syntactic constructions but punctuation)**.

**Page 23 top sentence**: *An important aspect of this hypothesis and hypothesis 6 is...* this IS hypothesis 6! **Get rid of " and hypothesis 6"**.

**Figure 1**: Last sentence:
*In these ways, trees act as chemical*
*stirring agents that remove nutrients from the rooting depth and return them to the top of the soil and act as mechanical stirrers*
*moving material from depth to the land surface and then downhill.*
This sentence runs on a bit, and it implies that it is primarily bioturbation that moves material downhill rather than gravity. Surely bioturbation increases the likelihood of gravity-driven transport of material (erosion) further down the slope? **Consider rewording this sentence, or at least the last clause.**

**Figure 2c**. The green arrow labeled Nutrient Acquisition points into soil and not up the tree, showing nutrient redistribution rather than aquisition. **Either label the arrow as such, or "nutrient acquisition and redistribution" with a second green arrowhead pointing up the tree, or, if the label is left as it is, the arrow should only point up the tree.**

**Figure 4**. Technically "mycorrhizae" refers to the tree-fungus symbioses rather than the fungi themselves. **Better to label the hygroscopic water as "fungi?", or possibly as "mycorrhizal fungi?"**, with question marks, as I am sure there are also limits beyond which fungi cannot access tightly-bound water and we don't know those limits.

Table 1. The following sentence:
*These fungi colonize inside the plant cell of absorptive roots and are most noted for their ability to improve phosphorus acquisition and other relatively immobile nutrients.* **should be reworded so that "acquisition" applies to the other relatively immobile nutrients as well as P, for example: "...*their ability to improve acquisition of phosphorus and other relatively immobile nutrients.*"**

---

## Author Response (AR2)

Dear Editor:

Thank you for the deadline extension. The authors have applied the recommendations as follows:

Review report #2.

We revised section 4 explicitly to include discussions of hypotheses, but mostly to tie hypotheses together. We also took Figure 1 and added the hypotheses to the figure both to strengthen that figure and to help us in Section 4. We think it helps to see the hypotheses listed somewhere, as suggested by the reviewer. We also revised Section 5 a little bit, as described below.

Final minor corrections:

We fixed the space on page 2, line9.

We added terrestrial as requested by the reviewer, and fixed the dash issue throughout manuscript.

We got rid of the extra "and hypothesis 6".

We reworded and clarified the last sentence in caption of Figure 1.

We revised Figure 2c as requested.

We revised the water figure (figure 4) to make text bigger and clearer, and we added in some of the suggestions from the reviewer.

Table 1...we revised the sentence as suggested by the reviewer for clarity.

Co-editor comments

We revised the abstract without making it longer so that it did not end so abruptly.

We are now using Figure 1 both as the intro and summary of hypotheses and we think this helps the paper a lot.

We added in a few references that are more general, about effects of vegetation.

We revised 2/3 to a range of values from the literature, and we cited the paper by Schlesinger and Jaschko, as suggested.

We made "some depth" more specific wherever we used the phrase.

p16. Revised as requested by reviewer

p23. We have tried to now clarify this indicating that the paper is about grassland.

p 23 We have now further emphasized the 2WW hypothesis is controversial (it was controversial among the authors of the paper!) and added the two refs that were requested. We had thought the original text made it clear 2WW is controversial but we tried to strengthen this.

p 25 We cite the Martinez-Vilalta paper in the manuscript now, but not precisely where the reviewer requested. We added another reference for the point about universal wilting point.

p 30. In the Conclusion section, we added in the url's for all of the databases as suggested by the reviewer. We didnt know about these and we think this is terrific.

We think the paper is much improved and we appreciate the reviews.

We hope the editor will invite us to upload the final version of the paper. Thanks Sue

[revised manuscript text omitted]